# Grain boundary mediated hydriding phase transformations in individual polycrystalline metal nanoparticles

Svetlana Alekseeva[1], Alice Bastos da Silva Fanta[2], Beniamino Iandolo[2,5], Tomasz J. Antosiewicz [1,3], Ferry Anggoro Ardy Nugroho [1], Jakob B. Wagner[2], Andrew Burrows[2], Vladimir P. Zhdanov[1,4] & Christoph Langhammer [1]

Grain boundaries separate crystallites in solids and influence material properties, as widely documented for bulk materials. In nanomaterials, however, investigations of grain boundaries are very challenging and just beginning. Here, we report the systematic mapping of the role of grain boundaries in the hydrogenation phase transformation in individual Pd nanoparticles. Employing multichannel single-particle plasmonic nanospectroscopy, we observe large variation in particle-specific hydride-formation pressure, which is absent in hydride decomposition. Transmission Kikuchi diffraction suggests direct correlation between length and type of grain boundaries and hydride-formation pressure. This correlation is consistent with tensile lattice strain induced by hydrogen localized near grain boundaries as the dominant factor controlling the phase transition during hydrogen absorption. In contrast, such correlation is absent for hydride decomposition, suggesting a different phase-transition pathway. In a wider context, our experimental setup represents a powerful platform to unravel microstructure–function correlations at the individual-nanoparticle level.

---

[1] Department of Physics, Chalmers University of Technology, Göteborg 412 96, Sweden. [2] Center for Electron Nanoscopy, Technical University of Denmark, Fysikvej, 2800 Kgs Lyngby, Denmark. [3] Centre of New Technologies, University of Warsaw, Banacha 2c, Warsaw 02-097, Poland. [4] Boreskov Institute of Catalysis, Russian Academy of Sciences, Novosibirsk 630090, Russia. [5]Present address: Department of Microtechnology and Nanotechnology, Technical University of Denmark, Ørsteds Pl., 2800 Kgs Lyngby, Denmark. Correspondence and requests for materials should be addressed to C.L. (email: clangham@chalmers.se)

Phase transformations in solids are of broad technological relevance in various applications including fabrication of metal alloys[1, 2], and also in numerous other contexts such as lithium ion batteries[3, 4] and hydrogen storage systems[5]. Mechanistically, such transformations typically are induced by sorption of solute atoms that results in the decrease of free energy. In such processes, grain boundaries are very important because they may enhance diffusion inside a polycrystal[6], act as sinks for the accumulation of impurities or segregated elements due to their different energetics[7, 8], and serve as mediators for plastic deformation[9, 10]. The corresponding experimental and theoretical studies focused on bulk materials can be tracked for many decades. In nanomaterials science, however, this subject is very little explored for two reasons. First of all, in very small nanoparticles the particle structure tends to relax to the single-crystal state. Second, for larger structures, where polycrystallinity is more likely, the investigation of grain boundaries is very challenging, despite the availability of high-resolution electron microscopy[11], X-ray imaging, and diffraction[12–14] techniques.

A particular field where grain boundaries are expected to be important mediators for both thermodynamic[15, 16] and kinetic[6] properties is hydrogen in metals and hydride formation. This phenomenon is traditionally associated with hydrogen embrittlement[17] and hydrogen storage[5], as well as the use of metal hydrides as hydrogen sensors[18]. To this end, the interplay between lattice coherency strain and dislocation nucleation in the particle-size dependence of hydride formation has recently been investigated for single-crystal Pd nanoparticles[11, 12, 14, 19–22].

Here we address a different but equally important aspect of solute-induced phase transitions by mapping out the role of grain boundaries and grain size in polycrystalline nanoparticles with nanosized grains, at the single-nanoparticle level. In such systems, grain boundaries are expected to be of significant importance due to the relative abundance of grain boundary sites compared to bulk materials with larger grains[16, 23–27]. We employ a multi-channel variant of plasmonic nanospectroscopy[20], which enables measurements of the individual response from up to 10 nanoparticles simultaneously, during both hydrogen absorption and desorption, and combine it with transmission electron microscopy (TEM) and transmission Kikuchi diffraction (TKD)[28]. This is an advance compared with the state of the art[11, 14, 19, 20, 22, 29, 30], where only sequential measurements of individual nanoparticles are possible, meaning that artifacts due to measurement-to-measurement variations cannot be avoided. At the same time we also highlight that in plasmonic nanospectroscopy, the obtained information is spatially averaged over the entire particle, in contrast to the recent electron energy loss spectroscopy and X-ray studies, where the hydride-formation process can be spatially resolved inside a single nanoparticle[11, 14].

Employing this platform, we reveal the details of Pd nanoparticle-grain-boundary structure, type and orientation, and find correlation between length and type of grain boundaries in individual nanoparticles, and their hydride-formation pressure. Using an analytical model, we identify tensile lattice strain, induced by hydrogen atoms near grain boundaries, as the main factor controlling the phase transition during hydrogen absorption. This indicates that polycrystalline nanoparticles can be understood as agglomerate of single crystallites that exhibit similar characteristics to "free" nanocrystals, whose interaction is mediated by the grain boundaries.

## Results

**Plasmonic nanospectroscopy–electron microscopy correlation.** Electron back-scatter diffraction (EBSD) is often the first choice when grain boundary microstructure characterization of materials is required[31–34].

However, it lacks the spatial resolution necessary for studying nanocrystalline materials[35–37]. Since the recent seminal work of Keller and Geiss on the sister technique TKD[28], this limitation has been ameliorated, and a spatial resolution on the scale of 5 nm can be achieved[38]. Furthermore, TKD does not require flat bulk specimens, as it is the case for EBSD, which enables the

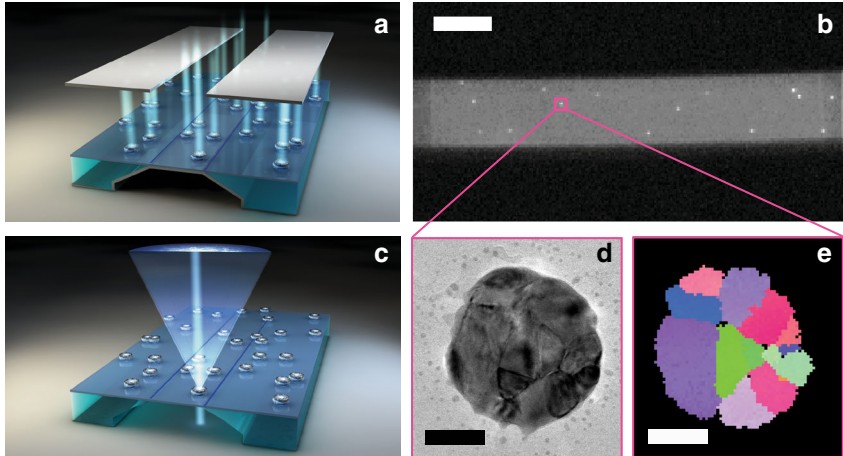

**Fig. 1** Experimental platform. Plasmonic nanospectroscopy is combined with transmission Kikuchi diffraction (*TKD*) and transmission electron microscopy (*TEM*). **a** Artist's rendition of the TEM-window platform with a Cr mirror layer grown beneath the membrane. This mirror layer enables multichannel single-particle plasmonic nanospectroscopy based on enhanced visible light back-scattering under dark-field illumination conditions, using the slit of a spectrometer (indicated as the *two dark rectangles*) to select the desired sample area. **b** CCD image through the spectrometer slit, showing a set of individual Pd nanoparticles as *bright dots* on the TEM membrane with Cr mirror underneath. Employing the slit to select a region on the sample, it becomes possible to image and spectroscopically address up to 10 single nanoparticles simultaneously, provided all are well-separated along the slit axis. The *scale bar* is 20 μm. **c** Artist's rendition of the TEM-window platform in "electron microscopy mode" after removal of the Cr mirror layer to enhance electron-transparency of the specimen. In TKD, using a state-of-the-art Bruker OPTIMUS^TM detector, the electron beam enters the specimen from the backside and the detector collects the Kikuchi diffraction pattern in transmission mode. **d** TEM image and **e** corresponding TKD grain orientation map of a single Pd nanoparticle (see Fig. 5 for corresponding grain orientation legend), which can be directly correlated with the plasmonic nanospectroscopy experiment by identifying the same nanoparticles in both experiments. The *scale bar* is 50 nm

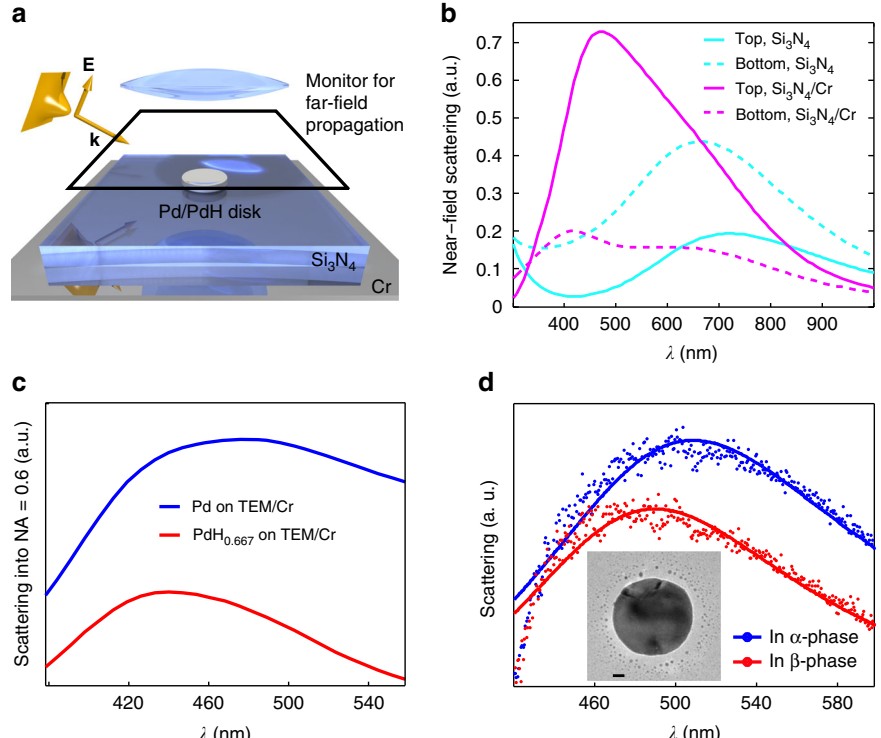

**Fig. 2** Finite difference time domain (FDTD) simulations. **a** Artist's rendition of the simulation scheme, where a Pd/PdH$_x$ nanodisk is placed on top of a 40 nm silicon nitride layer with a 10 nm Cr mirror below. A broadband Gaussian beam with TM polarization is incident onto the disk at 55° to mimic the dark-field illumination conditions of the plasmonic nanospectroscopy experiment. A combined scattered and reflected signal is collected above the structure and the Gaussian source, and then propagated into the far field, where it is analyzed. The reflected Gaussian beam is subtracted to obtain the scattered signal, which is integrated in the conical angle set by the numerical aperture of the collection lens and constitutes the "measured" signal in simulations. **b** Calculated backward and forward-directed scattering from a Pd nanodisk (diameter 140 nm, thickness 30 nm) placed on a TEM window in presence and absence of the Cr mirror layer underneath the membrane. The placement of the Cr mirror increases the backward/forward scattering ratio by a factor of ca. 6. **c** Calculated backward scattering into an objective with N.A. = 0.6 of a Pd nanodisk in the metallic (*blue line*) and fully hydrided (*red line*) state located on a TEM window with Cr mirror layer. **d** Corresponding experimentally measured dark-field scattering spectra of a Pd nanodisk located on a TEM window with Cr mirror layer in the metallic (*blue line*) and fully hydrided (*red line*) state. The *scale bar* in the *inset* TEM image of the Pd nanodisk is 20 nm. See Supplementary Discussion for further details

investigation of nanoparticles (Supplementary Fig. 1). However, TKD has yet to be explored in this field (Supplementary Discussion). Like TEM, it requires an electron-transparent specimen, as experiments are done in a transmission configuration. Conventional TEM grids or membranes are, however, problematic for plasmonic nanospectroscopy because of the relatively small grid spacing or etched cavities in the membranes, which cause stray light scattering or block some of the incoming/scattered light at the low angles that are typically required.

To overcome these limitations, we have developed a solution based on TEM "windows" that consist of a 40 nm electron-transparent amorphous Si$_3$N$_4$ membrane[39], combined with the reversible physical vapor deposition of a 10 nm Cr layer on the backside of the membrane, to enable TKD and plasmonic nanospectroscopy from the same sample (Fig. 1). As finite difference time-domain (FDTD) simulations of the light scattering by a single Pd nanodisk reveal, the Cr layer acts as a mirror that creates interference effects (Fig. 2a, b; Supplementary Figs. 2 and 3). These effects enhance the intensity of light back-scattered from the Pd nanoparticles (which generally are poor scatterers[40]), and thus make them visible on a CCD-chip. After the experiment, the Cr mirror layer is removed by wet etching from the backside (assuring that the etch does not interact with the nanoparticles) to transform the sample to its electron microscopy compatible state. This facilitates TKD and TEM analysis on the same single

nanoparticles as those probed in operando by the plasmonic nanospectroscopy. The cycle can be repeated multiple times by subsequently growing and removing the Cr layer.

**Plasmonic nanospectroscopy of hydrogen sorption.** Plasmonic nanospectroscopy of hydrogen sorption relies on the fact that the localized surface plasmon resonance (LSPR) frequency of a hydride-forming metal nanoparticle is proportional to the hydrogen uptake throughout the $\alpha$-phase region at low-hydrogen partial pressure where hydrogen is diluted at low concentration in a solid solution; the $\alpha + \beta$-phase-coexistence region ("plateau") at the first-order phase transition to and from the hydride ($\beta$-phase); and finally, the pure $\beta$-phase region at high-hydrogen partial pressure[18, 41, 42]. For our specific system, this concept has been corroborated by employing again FDTD simulations, which predict a distinct decrease of the Pd nanoparticle LSPR peak-scattering intensity, $\Delta$PI, and a spectral shift, $\Delta\lambda_{max}$, of the LSPR peak to the blue upon transition from the metallic to the hydride state (Fig. 2c). This agrees well with the experimentally observed response of a single Pd nanodisk upon hydride formation (Fig. 2d) and thus confirms that tracking the $\Delta$PI or $\Delta\lambda_{max}$ signal makes it possible to record optical pressure-composition p-C isotherms of single nanoparticles. As these two signals are proportional (Supplementary Fig. 4), we will use these two readouts interchangeably.

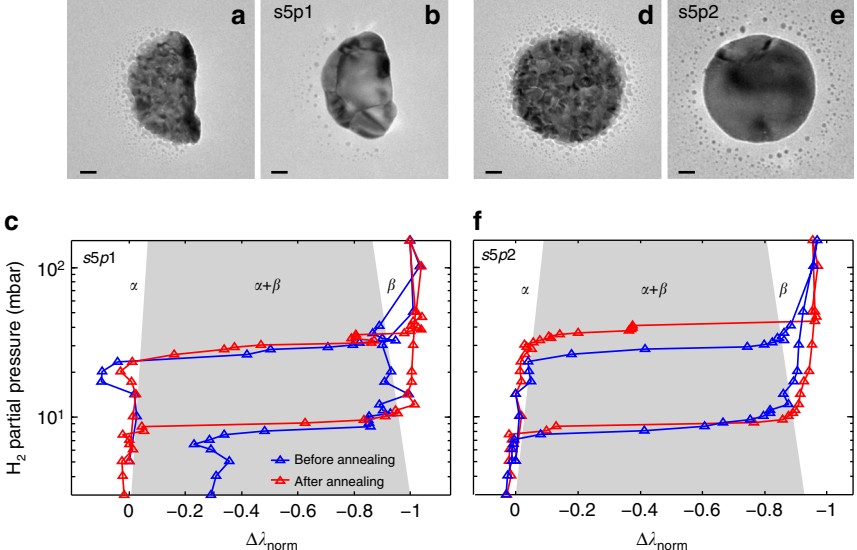

**Fig. 3** Correlated TEM and multichannel plasmonic nanospectroscopy. TEM micrograph of a Pd nanoparticle **a** directly after Pd evaporation through the mask used for nanofabrication[57] and **b** after annealing at 553 K for 12 h in Ar (in its designation, s5p1, the first number indicates the data set, s, and the second one the number of the particle, p, in this data set). Observe the transition from a state characterized by distinct polycrystallinity with very small grains on the order of 5–10 nm, to a polycrystal comprised of six grains of much larger and comparable size. The small "satellite" features around the nanoparticle are formed during the nanofabrication. Owing to their small size, they do not contribute to the measured signal. **c** Corresponding optical pressure-composition (p-C) isotherms measured before and after the annealing, revealing very similar properties with almost identical plateau pressures for hydride formation and slope of the $\alpha + \beta$-coexistence plateau. The hydrogen uptake (*horizontal axis*) is characterized by using the normalized wavelength shift signal, $\Delta\lambda_{norm}$. **d–f** Same as **a–c** for a second nanoparticle (s5p2) measured simultaneously. The annealing transformed this particle into a polycrystal that comprises two grains only, with very different size. This is directly reflected in the corresponding p-C isotherms, which reveal a distinct increase in hydride-formation plateau pressure in combination with the appearance of a double-plateau feature. This can be understood by considering that the two grains transform to the hydride phase at different pressures due to their distinctly different size. Notably, desorption plateau pressures are close to identical for both particles and unaffected by the change in microstructure. A version of **c** and **d**, which includes *error bars* along the $\Delta\lambda_{norm}$ axis is shown in Supplementary Fig. 10. The *scale bar* in the TEM images is 20 nm

**Correlating grain structure and (de)hydrogenation isotherms.** The microstructure of two as-grown Pd nanoparticles revealed by TEM is characterized by distinct polycrystallinity with very small grains on the order of 5–10 nm or smaller (Fig. 3a, d). These transform into different microstructures after 12 h annealing at 553 K in Ar, i.e., a polycrystal that comprises six individual grains (Fig. 3b) and a polycrystal that comprises two grains with significantly different size (Fig. 3e). We inspect the corresponding optical p-C isotherms obtained simultaneously for both particles before and after annealing. For example, the particle composed of six grains shows almost identical pressures for hydride formation and decomposition (Fig. 3c). In addition, we observe that its hydrogenation plateaus are sloped, most likely due to different hydrogenation pressures of the grains[11]. This is in contrast to the second particle, which exhibits a distinct increase in pressure, and split of the hydride-formation plateau, $P_{abs}$, from 27 to 37 and 42 mbar, respectively, due to the appearance of a double-plateau feature (Fig. 3f). This can be understood as that for a nanoparticle comprised two crystallites with significantly different size, each crystallite transforms to the hydride phase at different pressure, which is dictated mainly by size-dependent coherency strain at the interface between the $\alpha$- and the hydride phase[21]. In contrast, the desorption plateau pressures, $P_{des}$, are essentially identical for both particles and surprisingly unaffected by annealing and corresponding change in microstructure, indicating a different phase transformation pathway.

The asymmetry in structure sensitivity between hydride formation and decomposition becomes even more striking when looking at a large data set obtained from 32 thermally annealed nanoparticles on the same sample (Fig. 4 and Supplementary Figs. 6–10 for raw data and Supplementary Figs. 14 and 15 for

isotherms); these data points were obtained in groups of up to 10 individual nanoparticles probed simultaneously). Although hydride decomposition occurs within ca. 3 mbar (close to experimental uncertainty) for all investigated particles, the spread in hydride-formation pressure within a single and —importantly —simultaneously measured data sub-set is as large as 20 mbar. It can also be seen that multiple plateaus appear for several nanoparticles upon hydride formation (some of which are due to the simultaneous measurement of several nanoparticles with different plateau pressures; Supplementary Figs. 11–14). The single-particle data also agree well with corresponding ensemble measurements on arrays of Pd nanoparticles nanofabricated in the same way (Supplementary Fig. 18). Specifically, the characteristic plateau slope observed in hydride formation and decomposition isotherms obtained from nanoparticle ensembles (indicated as *green-shaded areas* in Fig. 4) corresponds to the spread in plateau pressures observed for the individual nanoparticles constituting the ensemble.

Asymmetric changes in the hydrogenation hysteresis of Pd thin films, observed in systems adhering strongly to the support, have been explained by the asymmetric interplay (compensation effect) between elastic compression induced by clamping (symmetric reduction of hysteresis) and energy loss due to plastic deformation (increases hysteresis), which enhance each other during hydride formation, and counteract each other upon desorption[43]. As the Pd nanodisks in the present case adhere to the support (despite the lack of adhesion layer, we see no sign of buckling or degradation of the particles upon cycling), a similar explanation of the observed asymmetry could be put forward. However, almost perfect compensation of compressive stress and plastic deformation effects is highly unlikely and, most importantly, we

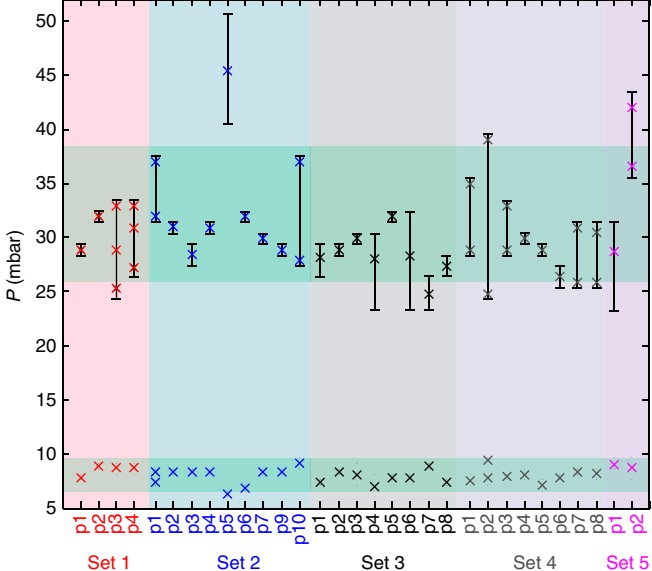

**Fig. 4** (De)hydrogenation pressures of single Pd nanoparticles. The $\alpha + \beta$ equilibrium plateau pressures for hydrogen absorption (*crosses* in the *upper part*) and desorption (*crosses* in the *lower part*) were determined from the corresponding hydrogenation traces (Supplementary Figs. 6–10) measured at 303 K for different single particles in data sub-sets s1, 2, 3, 4, and 5. The measurements on the particles in each sub-set were performed at the same time using multichannel single particle plasmonic nanospectroscopy. Sub-set 1 and 2 were located on the membrane region of the sample, whereas sub-set 3 and 4 were located on the bulk-area of the TEM-window substrate, adjacent to the membrane area (Supplementary Fig. 5). Sub-set 5 was on a separate sample. Particles in sub-sets s1–4 were annealed for 12 h at 743 K in Ar, and the particles in sub-set 5 were annealed for 12 h at 553 K in Ar. The *green-shaded areas* indicate the plateau slope observed in hydride formation and decomposition isotherms obtained from nanoparticle ensembles nanofabricated using the same procedure (Supplementary Fig. 18). Clearly, the slope coincides with the spread in plateau pressures observed for individual nanoparticles. The *error bars* (or *small dots* for desorption) represent the difference in pressure between the low and high pressure end of the plateau at the phase transition in the single-particle experiments [20]. However, in cases where the plateau spans directly between two data points, the true plateau width is expected to be lower and thus not resolved in our experiment (the single-crystalline particle s2p5 is a good example)

have also observed the same asymmetry in structure sensitivity of the two hysteresis branches for single-crystalline colloidal nanoparticles of different size and shape, which are not clamped by the support[20]. Therefore, we argue that the reason for the asymmetry is related to not only the thermodynamics but, even more importantly at the present relatively low temperatures, to the kinetics of dislocation formation. For example, the metal atom mass-transport during plastic deformation is expected to be different in the metal and hydride phases, due to, for example, spatial constraints induced by the presence of hydrogen atoms in the hydride phase and/or the difference in spatial localization of the hydride–gas, metal–gas, and hydride–metal interfaces. Consequently, dislocation formation is expected to be governed by different kinetics during hydride formation and decomposition, constituting a reason for the observed asymmetry. However, at present the understanding of dislocation formation kinetics in general, and especially in nanoparticles and during hydride formation/decomposition, is very limited[44–46], preventing a more rigorous and quantitative analysis beyond the recent work by Griessen et al.[21], which is in good agreement with our data.

**Role of grain boundaries in hydriding phase transformations.** To investigate the origin of the single-nanoparticle-specific hydrogenation characteristics identified in Fig. 4, we show in Fig. 5 a selection (the remaining isotherms are presented in Supplementary Figs. 12 and 13) of optical single particle p-C isotherms from data sets 1 and 2, together with corresponding TEM images and TKD grain orientation maps. Furthermore, high-resolution TEM images reveal lattice fringes for each nanoparticle (Supplementary Fig. 17), in agreement with columnar grains stretching from the substrate through the entire particle. However, they also show that the disk surfaces are not terminated with well-defined large facets with a specific orientation. From TKD, significant differences in microstructure characteristics become apparent at the individual-nanoparticle level, both in terms of number of grains, grain size, and grain orientation. For example, in data set 2, nanoparticle s2p2 is a polycrystal with 12 grains, whereas s2p5 is a single crystal. Comparing their p-C isotherms reveals significantly wider hysteresis for the single crystal. We therefore employ the unique opportunities offered by TKD to further characterize the microstructure of the individual nanoparticles by extracting quantitative descriptors such as total number of grains, average grain size, grain boundary length and fraction of high-angle grain boundaries (HAGB—defined as grains with lattice orientation mismatch >15°), and of twin boundaries. We then correlate these data with the information obtained from the plasmonic nanospectroscopy.

As the first descriptor, we chose the number of grains present in a nanoparticle as identified by TKD and plot it for all nanoparticles included in data sets 1 and 2 as a function of the hydride-formation plateau pressure (Fig. 6a). For comparison, we also include one data point from an unannealed Pd nanodisk (s5p2) as an example of the limit where grain boundaries and defects are highly abundant (grains of 10 nm or smaller). We find a clear correlation between number of grains and plateau pressure, $P_{abs}$, with the single crystal (particle s2p5 in Fig. 5) exhibiting the highest $P_{abs}$ value. For a larger number of grains $P_{abs}$ appears to asymptotically approach a value on the order of 25 mbar. This trend is even more accentuated when plotting $P_{abs}$ vs. the total grain-boundary length present in the nanoparticles (Fig. 6b).

**Grain-boundary tension.** To rationalize the results obtained so far, we recall that hydride formation in single-crystalline Pd nanoparticles can be understood in terms of hydrogen–hydrogen interactions, lattice strain, surface tension, sub-surface hydrogen, and the energetics of dislocation formation[14, 19–22]. These factors are significant for particles in the 1–1000 nm regime, very much depending on the specific effect. In bulk polycrystalline Pd, the grain size is typically much larger, and therefore an influence of grain boundaries on hydrogen sorption isotherms is not well manifested, except that hydrogen diffusion has been reported to be influenced[6, 16] and that for nanosized grains the phase-coexistence region has been observed to shrink[23, 24].

Focusing on polycrystalline nanoparticles, in analogy with the particle surface and sub-surface regions of a single crystal, grain boundaries possess boundary tension that induces intrinsic lattice strain and provides energetically favorable sites for hydrogen to occupy. Under comparable conditions, the grain-boundary tension is expected to be smaller than the particle size-related surface tension of a single-crystal nanoparticle because the metal–metal bonds at the grain boundaries are more saturated. In contrast, the number of energetically favorable sites (per unit area) for hydrogen at grain boundaries, and thus their relative importance, is larger (due to the specifics of their structure[47]) than at the nanoparticle surface because the surfaces of

nanoparticles are predominantly terminated by (111) facets, whereas on grain surfaces more open facets are more abundant. Moreover, the internal grain surface area in a polycrystalline nanoparticle is significantly larger than the external surface area, further highlighting the significance of grain boundary strain.

Accepting that this is the case, we focus on the effect of hydrogen occupying the sites near grain boundaries on the

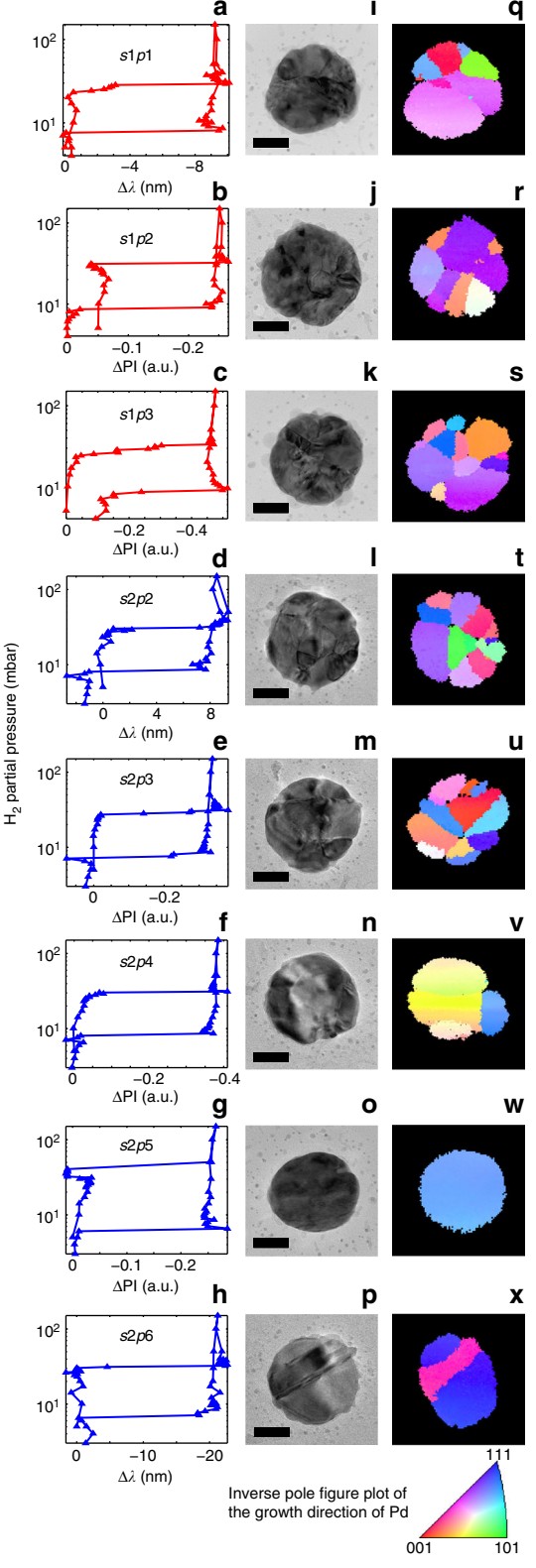

**Fig. 5** Single particle isotherms with TEM and TKD micrographs. **a–h** Optical p-C isotherms measured simultaneously by multichannel single particle plasmonic nanospectroscopy for a selection of single Pd nanoparticles. The data were obtained in two separate sub-set experiments, s1 (*red color* code) and s2 (*blue color* code). **i–p** TEM images of the corresponding Pd nanoparticles, which reveal distinct differences in microstructure with respect to number of grains and grain size. The *scale bar* is 50 nm. **q–x** Corresponding TKD grain orientation maps of the same nanoparticles reveal a single crystal (particle s2p5), a polycrystalline state with two or three large grains (s2p4 and s2p6), and a polycrystalline state with a multitude of small grains, in the same data set. The *color code* of the TKD images is explained in the inverse pole figure plot, which depicts the grain crystallographic orientation with respect to the out-of-plane axis. An equivalent plot using the alternative plasmonic nanospectroscopy readout parameter is shown in Supplementary Fig. 11

hydrogen adsorption isotherms. In analogy with surface and sub-surface sites of a single-crystalline particle, the grain boundary sites are energetically favorable and, accordingly, they are already occupied in the phase-coexistence region and generate tensile lattice strain. Consequently, the location of the coexistence region along the hydrogen-pressure axis (i.e., $P_{abs}$), is shifted to lower hydrogen pressures for increasing amount of grain boundary per volume. Therefore, also in analogy with the description for sub-surface sites, the corresponding shift of the chemical potential of hydrogen atoms occupying regular lattice sites can be estimated analytically (Supplementary Discussion and Supplementary Fig. 19). In particular, we can calculate an equilibrium plateau pressure reduction factor for hydride formation, $f_{2D}$, as a function of average grain radius. As shown in Fig. 6c, this predicts a reduction of the equilibrium pressure by ca. 30% in a grain radius range from 80 to 20 nm, which is in good agreement with the experimental data points plotted in the same graph. In addition, it is of interest that $P_{abs}$ for particles with ca. 40-nm grain diameter is very similar to that for ca. 30–40-nm-sized single-crystal Pd nanocubes[20]. This is in line with our arguments because the effect of hydrogen located near grain boundaries is comparable to that of hydrogen located in the sub-surface sites of the single-crystalline cubes. We thus conclude that tensile lattice strain induced by hydrogen absorbed near grain boundaries is an important mediator of the observed significant spread in hydride-formation equilibrium pressure of polycrystalline nanoparticles of the same size and shape.

This is further corroborated by plotting the hysteresis factor, $\ln(P_{abs}/P_{des})$, as function of the grain-boundary length (Fig. 6d). This factor is on the order of 1.2 for the particles with highest grain-boundary length, which is in good agreement with constraint-free "buckling" Pd thin films, where plastic deformation contributions to hysteresis are minor[43]. This indicates that the observed effects indeed are of elastic origin. To this end, we also note that the considered grain sizes are far larger than the size regime where inverse Hall–Petch strengthening, which in principle could account for the observed dependence if it were mediated by plastic deformation, is observed[48]. Finally, we highlight that the absolute value for the single-crystal particle (particle s2p5) is in excellent quantitative agreement with Griessen et al.'s coherent interface model[21], which further corroborates the dominant role of elastic lattice strain. In addition, our results shed light on the so far unexplored transition (cf. Fig. 3 in ref. [21]) from the size-dependent hysteretic behavior of single-crystalline nanoparticles with very high hydrogenation pressures in large crystals[14] to the much lower value found for polycrystalline bulk[49] (Fig. 6a, b).

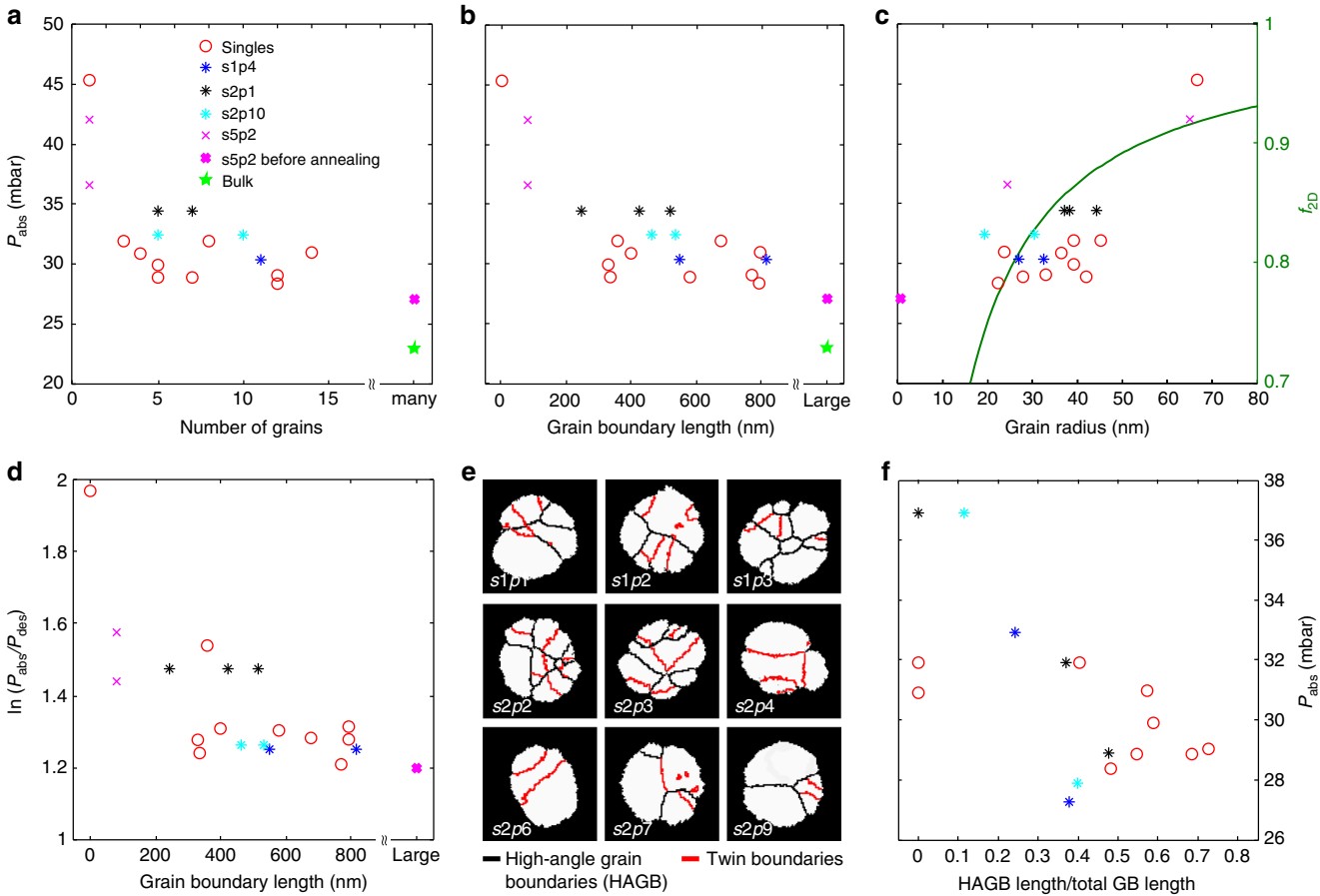

**Fig. 6** Grain structure–hydrogenation pressure correlations. The data forming the basis of this figure are summarized in Supplementary Table 1. **a** The equilibrium plateau pressure upon hydride formation measured by plasmonic nanospectroscopy plotted as a function of number of grains. The *red circles* correspond to data points obtained from single nanoparticles (cf. Fig. 5 and Supplementary Fig. 12). The *magenta crosses* correspond to the *disk-shaped particle* shown in Fig. 3d and e, before and after annealing (particle s5p2). The *blue, black,* and *turquoise stars* represent averaged plateau pressures for data points corresponding to multiple particles (cf. Supplementary Fig. 13), i.e., particles not resolved in the plasmonic nanospectroscopy experiment. The *green star* depicts the hydrogenation pressure for bulk Pd obtained by Lässer and Klatt[49]. We refer also to Supplementary Fig. 20 for an alternative representation of the same data. **b** Equilibrium hydrogenation plateau pressure as a function of grain-boundary length. **c** Hydrogenation plateau pressure as a function of average grain radius (left y axis). The grain radius was calculated from the TKD images by determining the area of a grain and then taking the radius of a circle with the same area. In the same graph, we also plot the theoretically calculated equilibrium plateau pressure reduction factor for hydride formation, $f_{2D}$, (right y axis) as a function of average grain radius. **d** Hysteresis factor, $\ln(P_{abs}/P_{des})$, as a function of grain boundary length. **e** TKD images of the same nanoparticles shown in Fig. 5, but with grain boundaries highlighted and categorized into high-angle grain boundaries (HAGBs—*black lines*) and twin boundaries (*red lines*). **f** Corresponding plot of HAGB fraction of total grain boundary length vs. equilibrium hydrogenation plateau pressure. The corresponding plots for absolute HAGB length and twin-boundary length are shown in Supplementary Fig. 21

Another aspect of grain boundaries that has been observed in nanocrystalline films[16, 23] is a characteristic narrowing of the miscibility gap. Inspection of our data in this respect does not reveal a significant correlation between grain-boundary length and width of the miscibility gap. We argue that the reason is the uncertainty of the $\Delta\lambda$ readout parameter, which is on the order of a few nm in the present case, and caused by the spectrally broad peak of the LSPR in Pd due to interband-damping[50]. This resolution is not enough to resolve this effect in the present regime of particles being comprised of 1–15 grains, where it is not expected to be very pronounced. However, we also note that, for particle s5p2 (Fig. 3f), where the change in microstructure from the before to after annealing state is much more drastic, there is a quite clear indication of a widening of the miscibility gap after annealing, when the sample comprises two grains only.

**Role of grain boundary type.** As the last step we further deepen our analysis by distinguishing two grain boundary types, i.e., twin boundaries (fulfilling the twin relationship of 60° lattice

misorientation about the (111) axis, which is most common in fcc materials) and HAGBs with lattice mismatch >15°. Low-angle grain boundaries are either completely lacking or <1% abundant and thus ignored. Figure 6e shows the corresponding TKD images for our single nanoparticles for data sets 1 and 2, with the grain boundaries highlighted and categorized in these two classes. Clearly, the relative abundance of the respective class varies widely from particle to particle. From a hydrogen sorption energetics point of view, this is relevant because twin boundaries are more close-packed than HAGBs and thus energetically more similar to bulk sites inside the grain. Hence, within the framework outlined above, the presence of twin boundaries is expected to influence the hydride-formation equilibrium pressure to a lesser extent than HAGBs. When plotting the HAGB length normalized by the total length of all grain boundaries, we indeed observe a trend towards larger reduction of the equilibrium pressure for the particles where the HAGBs are most abundant (Fig. 6f). Finally, we mention that a competing contribution of dislocation formation inside the individual (single crystalline) grains as

mediator for the observed variations of hydrogenation pressure is highly unlikely in view of the fact that our grains, with grain radii of 45 nm or below (Fig. 6c), are significantly smaller that the critical size for dislocation formation identified by Ulvestad et al[14].

Concerning the hydride decomposition, it occurs from the state when the entire lattice is occupied by hydrogen and when lattice expansion is caused primarily by hydrogen at regular interstitial sites inside the grain. Accordingly, the role of grain-boundary sites, and thus of grain boundaries, on $P_{des}$ is expected to be (much) less important. This may explain why the distribution of $P_{des}$ (Fig. 4) is very narrow and microstructure independent.

In summary, multichannel plasmonic nanospectroscopy is effective for probing up to 10 individual functional nanoparticles in situ and simultaneously. In this way, ensemble averaging and experiment-to-experiment uncertainty are eliminated, and unambiguous identification and quantification of single particle-specific effects are firmly possible. TKD enables characterization of microstructure and grain boundaries in structural materials in general and nanoparticles in particular, with nanometer resolution. We have combined these two techniques, together with TEM imaging, to investigate in detail, at the single-particle level, the role of grain boundaries in the hydrogen-induced phase transition during hydride formation and decomposition in a large set of identically sized polycrystalline Pd nanoparticles. As the main results, we find distinct asymmetry in the dependence of the hydride formation and decomposition equilibrium pressures on the microstructure of individual nanoparticles. The corresponding pressure for hydride formation directly correlates with grain-boundary length and grain-boundary type in each particle. In contrast, microstructure–phase transformation pressure correlation is absent for hydride decomposition. Using an analytical model, we identify tensile lattice strain induced by hydrogen absorption near the grain boundaries as the dominant factor controlling the adsorption branch of the phase transition. This finding also implies that, within each grain, the phase transition is coherent, that is, no sharp phase boundaries between $\alpha$-and $\beta$-phases exist. Thus, it also corroborates an earlier TEM study on coherent $\beta$-phase precipitation in Pd foil[51]. Furthermore our detailed TKD analysis shows that high-angle grain boundaries are the main contributor, and that twin boundaries are less important. The observed structure-insensitivity of the hydride decomposition, which we also have observed for single-crystalline nanoparticles[20], suggests a different phase-transition pathway, most likely via an incoherent unloading process involving plastic deformation, as recently proposed by Griessen et al.[21] Thus, our results indicate that polycrystalline nanoparticles during a hydriding phase transformation can be conceptually understood as agglomerate of single crystallites exhibiting similar characteristics to "free" nanocrystals, whose interaction is mediated by the grain boundaries. Moreover, they shed light on the transition from the size-dependent hysteretic behavior of single-crystalline nanoparticles to the much lower hydrogenation pressures observed for polycrystalline bulk.

In a wider perspective, we predict that our general approach can be used to scrutinize the role of grains and grain boundaries in essentially any metal hydride system based on the fact that numerous plasmonic sensing studies on ensembles of different hydride-forming metal nanoparticle systems already exist (e.g., AuPd alloys[52], Mg[53, 54], Y[55]). Furthermore, it can be easily expanded to other processes of interest in metallic nanostructures where oxidation and reduction are a prominent example. Owing to sizeable mismatch of the lattice spacing between metal and oxide, the formation of grains in the oxide is nearly inevitable and has long been expected to have a key role in oxidation/reduction

processes (see, e.g., ref.[56] and references therein). The underlying physics is, however, still far from clear, especially on the nm scale.

## Methods

**Sample preparation.** The Pd nanodisks were fabricated by hole-mask colloidal lithography[57] using a highly diluted ($10^{-4}$ wt%) polystyrene sphere (sulfate latex, Interfacial Dynamics Corporation, size 140 nm, evaporated Pd thickness 30 nm) solution and a short incubation time of 10 s, to achieve the low particle density necessary for plasmonic nanospectroscopy. The particles were fabricated on square, $150 \times 150$ μm, 40-nm-thick $Si_3N_4$ membranes supported by bulk silicon on all four sides[39]. To facilitate plasmonic nanospectroscopy from the membrane region, a 10 nm Cr mirror layer was electron beam evaporated on the backside of the TEM windows (evaporation rate of 1 Å/s in a Lesker PVD 225 Evaporator, base pressure $<5\times10^{-7}$ Torr). To transfer the sample into its electron microscopy compatible state, the Cr film was removed by applying Cr etch for 2 min (Sunchem AB, NiCr etchant 650095, composition: ceric ammonium nitrate 10–15%, nitric acid 15 −20%, DI water 60–70%) to the backside of the TEM window.

**Single particle dark-field scattering spectroscopy.** For the hydrogen sorption experiments, the samples were placed in a temperature-controlled and gas-tight microscope stage (Linkam, THMS600) that was connected to a set of mass flow controllers (Bronkhorst, Low-ΔP-flow and EL-flow) to supply the desired gas flow and concentration to the sample. We used Ar as carrier gas (6.0 purity) and mixed it at different concentrations with 100% $H_2$ gas (6.0 purity), and operated the system at atmospheric pressure. After identifying a set of particles aligned with the spectrometer slit (opened 1000 μm, Andor Shamrock SR303i) using an upright optical microscope (Nikon Eclipse LV100, Nikon 50 × BD objective) the light scattered from the particles was dispersed onto a grating (150 lines/mm, blaze wavelength 800 nm) from which it was analyzed by a thermoelectrically cooled CCD camera (Andor Newton 920). This limits the maximal number of particles possible to analyze simultaneously to something between 10 and 25 (the higher number could be achieved by using electron-beam lithography to nanofabricate particles aligned in a single row). However, employing concepts like hyperspectral imaging, significantly more particles can be analyzed simultaneously at the cost of significantly decreased data acquisition speed[58, 59]. The illumination source of the microscope was a 50 W halogen lamp (Nikon LV-HL50W LL). Normalized-scattering spectra $I_{sc}$ from individual nanoantennas were obtained as a function of wavelength $\lambda$ using the relation $I_{sc}(\lambda) = (S - D)/CRS$, where $S$ is the collected signal from an area with nanoantenna, $D$ is the signal from the nearby area without nanoantenna (dark signal for background correction taken from an area with identical pixel width but without particles), and $CRS$ is the signal collected from the diffuse white certified reflectance standard (Labsphere SRS-99-020). CRS is used to correct the signal for the lamp spectrum. The acquisition time for each spectrum was 10 s. Multiple spectra were collected simultaneously by using the Newton CCD camera in the multi-track readout mode. The obtained single-particle-scattering spectra were fitted with a Lorentzian function (±50 nm from the peak position) to derive information about the temporal evolution of the peak position and peak intensity. As was previously shown[42, 60], the changes in peak position are proportional to the hydrogen concentration in the probed Pd particle.

**TEM imaging.** Bright field TEM images of nanodisks were acquired from the "windows" mentioned earlier, using a Titan 80–300 TEM (FEI) operated at an accelerating voltage of 300 kV.

**TKD analysis.** The 40 nm silicon nitride TEM window with the Pd nanoparticles was mounted on a TKD sample clamp holder with the nanoparticles facing downward. The holder was installed on a FEI Nova Nano lab 600 stage and the sample was positioned in the microscope in horizontal position (0° tilt) at a working distance of 5 mm. The microscope was equipped with the recently introduced Bruker OPTIMUS™ TKD detector[61] and operated at an acceleration voltage of 30 kV and a beam current of 6.7 nA, using a 30 μm aperture. The detector was positioned in such a way that the smallest distance between the electron-beam focusing point at the specimen surface and the camera was 15 mm. All measurements were performed in low vacuum mode with a water vapor pressure of 50 Pa using a low vacuum detector placed at the microscope pole piece. Low vacuum was chosen to reduce sample drift. The TKD orientation maps were collected for each particle with a pattern resolution of $800 \times 600$ pixels ($2 \times 2$ binning), exposure time of 20 ms and step size of 3 nm. Before the data analysis, the raw data were processed to remove uncertain data points and to define a grain. A grain was defined as an area containing at least 3 data points with a mis-orientation larger than 5° with respect to its neighbor. All data sets containing <3 points were removed from the raw orientation map. Data analysis was performed using CrystAlign Bruker and OIM TSL softwares.

**FDTD simulations.** Calculation of the scattering spectra was done using FDTD Solutions from Lumerical, Inc. A Pd or $PdH_{0.667}$ disk (140 nm diameter, 30 nm thick) was placed on top of a 40 nm substrate with refractive index 2. Where relevant, a bottom 10 nm Cr layer was added. The permittivities of metals were

taken from Silkin et al.[62] and Palik[63] for Pd/PdH$_x$ and Cr, respectively. The structure was illuminated by a broad Gaussian beam incident at 55° (transverse-magnetic (TM) polarization so both longitudinal and transverse resonances of the disk can be excited), an angle within the illumination angle band of the dark-field setup. The beam's axis coincided with the center of the disk. Scattered and reflected light was collected across a wide monitor for further propagation into the far field to account for the numerical aperture of the collection objective (NA = 0.6). The scattered light was obtained by subtracting the light reflected from the appropriate substrates. The mesh around the disk and in the substrates directly below was refined to a size of 1 nm what assured converged results.

Additional calculations were conducted for a plane wave incident source with scattering measurements taken in the intermediate field ca. 150 nm from the geometrical center of the disk.

**Data availability**. The data that support the findings of this study are available from the corresponding author upon request.

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

## Acknowledgements

We acknowledge financial support from the ERC StG 678941"SINCAT" (C.L.), the Knut and Alice Wallenberg Foundation Project 2015.0057 (C.L.), the Chalmers Areas of Advance Nanoscience and Nanotechnology (S.S. and C.L.), the Swedish Foundation for Strategic Research Framework Program RMA11–0037 (F.A.A.N.) the Polish National Science Center via the project 2012/07/D/ST3/02152 (T.J.A.) and the Russian Federal Agency for Scientific Organizations (project 0303-2016-0001; V.P.Z.). The research leading to these results has also received funding from the People Programme (Marie Curie Actions) of the European Union's Seventh Framework Programme (FP7/2007–2013) under REA Grant Agreement No. 609405 (COFUNDPostdocDTU) (B.I.). We also gratefully acknowledge Bruker Nano GmbH (BNA) for the use of the OPTI-MUS<sup>TM</sup> TKD detector, P. Tabib Zadeh Adibi for the help with the first TEM characterizations, and Daniel Goran from BNA and I. Zorić for useful discussions.

## Author contributions

C.L. and S.A. planned the experiments, analyzed the data, and wrote the paper. S.A. nanofabricated the samples and performed the single-particle nanospectroscopy measurements. A.B., B.I. and A.B.d.S.F. designed the TKD experiments. A.B.d.S.F. performed the TKD experiments and analyzed the TKD data. B.I. and J.B.W. planned the TEM experiments. B.I. carried out the TEM experiments. F.A.A.N. performed the ensemble measurements. T.J.A. performed the FDTD simulations. V.P.Z. analyzed theoretically the effect of grain boundaries on absorption isotherms. C.L. conceived the general approach and coordinated the project.

## Additional information

**Competing interest:** The authors declare no competing financial interests.

