## [Peer Review File · Nature Communications]

Reviewers' comments:

Reviewer #1 (Remarks to the Author):

In this work Svetlana Alekseeva¹ et al. introduce a novel combination of characterisation of polycrystalline Pd-hydride nanoparticles. The paper is a solid piece of work, I cannot see major flaws or shortcomings on the experiments. A minor detail may be the statistical evaluation (Figure 4, a definition of the uncertainty of the plateau pressure would help here). The methods are interesting, with potential for future applications on similar metal-hydride system. However, this statement is related to my first critics: the paper entitles "metal nano particles" suggesting more than Pd, which is the potential, but has not been shown. Furthermore, apart from a demonstration of the techniques, the scientific claims are rather speculative, e.g., "suggest direct correlation between length and type of grain boundaries and hydride-formation pressure"; "The observed structure-insensitivity of the hydride decomposition, which we also have observed for single crystalline nanoparticles, suggests a different phase transition pathway, most likely via an incoherent unloading process involving plastic deformation, as recently proposed by Griessen et al." As the authors claim to have evidence for a quantitative correlation, a quantitative model should then corroborate this statement. It is not clear to me, why plastic deformation affect only the hydride formation.

Were the particles cycled? If not, the explanation may just be due to the initial barrier of defect formation. Finally, this study focuses on the hydrogen uptake by nano-particles. It is well known that in particular nanoparticles are sensitive to surface contamination. Although Pd is a relatively inert material, surface and also grain boundaries may be contaminated by carbon, and/or oxygen. In many materials, the stability of materials depends on the stability of the grain boundaries potentially weakened by contaminations (e.g., sulfur in steel).

In addition to the specific critics I miss some lines on future perspectives in the field. Pd hydride nanoparticles are one of the few systems, where such investigations are possible. If the authors see further potential of this method, this is the place to mention it!

Reviewer #2 (Remarks to the Author):

Overall, I am impressed by the both the quantity and depth of the experimental results. The investigation of the role of grain boundaries in the hydriding phase transformation is an open question and important for fundamental and technical reasons. However, I do have some concerns. I separated the concerns into major and minor.

Major

- All of the information is spatially averaged over the entire particle. This needs to be explicitly stated in the text, especially when comparing it to refs. 13, 16 where the hydride formation is spatially mapped within the particle. It also assumes that the grains are columnar, i.e. extend through the thickness of the nanoparticles. Is there evidence for this? What could be the signature if the grains are not columnar?
- There is no sensitivity to intragrain defects (particularly dislocations). It's known that dislocations play a very important role in the transformation pressure. Have the authors thought about how to exclude or account for the potential impact of dislocations within the grains?
- The data in Fig. 6 a-d shows basically no dependence of the transformation pressure on the independent variable being plotted if the initial and final points are excluded. For example, in Fig. 6b, from 200-800 nm the transformation pressure is effectively flat. I think the authors need to revise

their conclusions a bit here. It's not clear to me that there is a strong effect. Fig. 6f does a bit better job. So, it might be that the different types of grain boundaries are complicating their analysis. Still, this is only mentioned at the very end and needs to be discussed more thoroughly.

Minor:

- Line 23. "nearly lacking" is not a good word choice. Perhaps "beginning to be explored"
- Line 31-32. The technique is not sensitive to strain so "we identify tensile lattice strain" should be removed. A better sentence would be "The absorption pressure dependence we observe is consistent with tensile lattice strain..."
- Line 35-37. In my understanding, the technique will work for metals that have phase transformations where the plasmon conduction band changes by a lot. What are some examples of other systems where this is true?
- Line 67. References need to be added to the significant literature that exists for nano crystalline thin films, especially those dealing with nano crystalline palladium
- Line 71. The TEM image shows residual stuff around the particle. Is this Pd or something else?
- Line 71: "This is a significant advance compared to state of the art". I don't agree with how refs 13 and 16 are characterized. In Dionne et al. the hydride decomposition process is investigated (see Fig. 3 in "In situ detection of hydrogen-induced phase transitions in individual palladium nanocrystals). In Ulvestad et al., the decomposition process can be mapped as well. It is just that the diffraction signal is very complicated due to the high dislocation density in the particles that the authors did not do this. It is as simple as doing loading/unloading measurements with bulk ensemble x-ray diffraction if no imaging is performed.
- Line 73-74: "where a new experiment is necessary for each studied nanoparticle". This is not true. In Ulvestad et al. up to 25 particles were measured sequentially at each hydrogen partial pressure. Once the measurement of all particles was finished, the pressure was incremented and all particles were measured at the next pressure. Using the technique of Dionne et al., the same sequential approach can be used. No new experiment is required to measure multiple particles.
- Line 71-74: The main difference of this technique is the simultaneous measurement. But it should also be pointed out that this technique does not give spatially resolved information unlike refs 13 and 16.
- Line 103. Are there problems with these windows breaking during the Cr deposition and removal?
- Line 104. Is it possible to get CrH₂ forming under the experimental conditions?
- Lines 109-113. How is it ensured that the same particle is imaged in all of these steps? Are there fiducial markers on the sample holders used?
- Line 118. Spelling of localized is incorrect.
- Line 119. Since the technique is sensitive only to the surface plasmon, isn't it possible that the hydrogen-rich surface layer that forms before the transformation pressure could give the false signal that the entire particle is in the hydrogen-rich phase? The hydrogen-rich surface layer is well documented to form below the transformation pressure for the entire particle.
- Line 131. I think it needs to be said that this information is spatially-averaged
- Line 145. What are the error bars in Fig. 3c?
- Line 166. How is equilibrium defined? How long is spent at each hydrogen partial pressure?
- Line 166:168. Wording here is confusing. Several particles with different plateau pressures are equivalent to multiple plateaus for a single particle?
- Line 181. Why do the authors think the particles are strongly adhered to the support?
- Line 190. "scrutinize" is not the right word choice. Suggest "investigate"
- Line 194. What is meant by lattice fringes here?
- Line 221-222. The factors are significant from 1-1000 nm for the loading pressure (Griessen Fig. 3). 1-100 nm for the unloading pressure.
- Line 226. Add reference to "Narrowing of the palladium-hydrogen miscibility gap in nano crystalline palladium" by J. A. Eastman, L. J. Thompson, and B. J. Kestrel Phys Rev B 1993

- Line 245-247. What parameters are being fit in the model and how do the values compare to the fitted parameters of the Griessen model?
- Line 253-256. This seems like a strong conclusion given Fig. 6a-d showing a very weak dependence if the initial and final points are excluded.
- Line 474. What is "TM" polarization?
- Fig. 2 c-d. Why is there a mismatch in the peak locations for the computed and the measured figures? For example, in Fig. 2c the red curve has a maximum at about 440 nm, whereas the maximum in Fig. 2d is 490 nm.
- Fig. 3c: Why is there a positive shift in $\Delta\lambda$ right before the loading transformation? It is more noticeable in the blue curve than in the red. This can also be seen in many of the other isotherms, for example Fig. S12.
- Fig. 4: In my understanding, this plot is to show that there is a big variance in the loading pressure but not in the unloading pressure. Less data could be used to make this point and make the figure potentially easier to read.
- Line 516. These are not "crosses" they are "xs". A cross is like this +

Minor comments on the SI:

- Line 126-128. It's confusing to read that the simulation contains only the Cr mirror but then shows that the Cr mirror enhances the Pd signal as said in Line 105-107 in the main text.
- Line 199. I do not see from the plots that there is a lack of a clearly defined, narrow LSPR. This needs a bit more explanation.

Reviewer #3 (Remarks to the Author):

This communication reports on an original combination of in-situ and ex-situ techniques for the study of hydriding equilibrium of Pd nanoparticles, together with microstructural characterisation. The key aspect of this work is the use of a single TEM window for both in-situ (plasmonic nanospectroscopy) and ex-situ (TKD and TEM) analyses, enabling to identify and track multiple particles submitted to the same experimental conditions. The results bring an interesting picture of the hydriding mechanism of palladium, because they provide the missing link between macroscopic ensembles of crystallites - i.e. from the well-known bulk Pd to nanostructured materials such as thin films and nanoparticles - and single particles/crystallites. Depending on the microstructure of each individual particle - combined with the p-C isotherms measured by plasmonic nanospectroscopy - interesting conclusions were drawn regarding the impact of grain boundaries on the miscibility gap between the hydride α and β phases. Therefore, I think that this paper is suitable for publication in Nature Communications. However, I have some comments which should be addressed, notably concerning the decoupling of grain boundary effects from these of other defects, and the experimental error on the p-C isotherms.

Important general comments:

1) Reading this manuscript gives the impression that the authors measured the effect of the sole grain boundaries on the p-C isotherms, as if annealing the specimens up to 470°C did not affect other crystalline defects. Every crystalline defect has its distinct effect on the hydriding mechanism (see e.g. A. Pundt and R. Kirchheim, *Annu Rev Mater Res* 36 (2006) 555), and annealing activates defect recovery, not only from grain boundaries, but also from dislocations, vacancies, twins, impurities, etc. In my view, this should not be overlooked, especially when studying such nanomaterials with high defect densities. I recognise that, besides grain boundaries, the authors take twin boundaries as well into consideration in their analysis, but I am particularly curious about the dislocation density before and after annealing. Do the authors have such defect statistics already? Otherwise, could they measure it or extract it from their TEM data? If grain boundary effects are really overwhelming the

other effects, this should be put into numbers. Otherwise, the authors should clarify why they think other effects can be neglected, and/or state which hypotheses were made regarding the effect of other defects.

2) The same remark applies to surface sites. In Pd, surface sites are by far the most energetically favoured, and at room temperature, surface coverage is close to unity even at low H₂ pressure (see e.g. M. Johansson et al., Surf. Sci. 604 (2010) 718-729). I am puzzled when I read the argumentation on p. 10 lines 227-237, I am not sure to understand what the authors mean here. In particular, the sentence "In contrast, the number of energetically favorable sites (per unit area) for hydrogen at grain boundaries, and thus their relative importance, is somewhat larger (due to the specifics of their structure) than at the nanoparticle surface" makes little sense to me, because even if this is true, surface sites are filled in priority and close to saturation. This passage should definitely be reformulated for more clarity, and, in relation to previous comment, the idea that the effect of grain boundaries is the main effect observed here should be put into numbers (e.g. the double occurrence of the word "somewhat" in this paragraph is not convincing).

3) I have some interrogations regarding the experimental error related to the p-C isotherm measurements. I am surprised by the sentence on p. 7, lines 144-145 ("The particle composed of six grains shows almost identical pressures for hydride formation and decomposition", see also caption of Fig. 3c). With such an important microstructural change, one would expect a sloped and narrowed miscibility gap after annealing, as correctly noticed by the authors when they refer to Refs. 18 and 41 in their manuscript. Also, the double plateau feature identified on Fig. 3f is not very clear, at least by looking at the figure (it is more obvious on other isotherms though). The plateau split is identified at 37 and 42 mbar (p. 7, line 149). These values are quite close together, and make me wonder if they can really be resolved. The way the data points are spread in Fig. 4 - although the contrast between the absorption and desorption isotherms is very interesting - also makes me wonder if there is no artifact here, e.g. a higher error at higher pressures (plus the error bars on this graph are not real error bars). The authors should address in numbers the error on the plateau pressures and on the $\Delta \ln \text{norm}$ parameter derived from plasmonic nanospectroscopy. Maybe this could explain some unexpected results, e.g. why narrowed miscibility gaps cannot be resolved (there is indeed some scattering at the plateau borders on several isotherms in Figs. 4 and 5), and give more confidence in the author's analysis.

Specific comments:

1) Page 2, lines 22-23: "In nanomaterials, however, investigations of grain boundaries are very challenging and nearly lacking". I would remove "nearly lacking", I think this statement is too strong. This is indeed challenging, but more than just several studies can be found on materials exhibiting very similar microstructures in terms of grain size, twin and dislocation densities (especially thin films).

2) Page 2, line 27: Why are the p-C isotherm measurements limited to 10 simultaneous particles? This is a key limitation that should be explained in the experimental section.

3) Page 4, lines 67-75: The authors cite 7 references in a row and advertise their technique, claiming that both hydrogen absorption and desorption can be measured (let's call it claim A), and that more than 1 particle can be studied at the same time (claim B). It sounds like none of these references fulfill these claims, but it is not true (e.g. Ref. 21 makes claim A). The authors should separate this list of references between the ones that satisfy claims A and B in order to better isolate the innovative aspects of their work.

Reviewer #1 (Remarks to the Author):

Q1: In this work Svetlana Alekseeva¹ et al. introduce a novel combination of characterisation of polycrystalline Pd-hydride nanoparticles. The paper is a solid piece of work, I cannot see major flaws or shortcomings on the experiments. A minor detail may be the statistical evaluation (Figure 4, a definition of the uncertainty of the plateau pressure would help here).

Our response: As we describe in the caption of Figure 4: "The error bars (or small dots for desorption) correspond to the obtained width of the plateau along the pressure axis". In other words the error bars define the lower and upper bounds of the "plateau" and thus the uncertainty of the exact plateau pressure.

Q2: The methods are interesting, with potential for future applications on similar metal-hydride system. However, this statement is related to my first critics: the paper entitles "metal nano particles" suggesting more than Pd, which is the potential, but has not been shown.

*Our response: We agree with the reviewer that this specific method has not been shown for another metal. However, numerous other hydride forming metal systems have been characterized using nanoplasmonic sensing, including PdAu alloys (ensemble - Wadell, C., et al. **Nano Letters**, 15, 3563-3570 (2015)), Mg (single particle - Shegai, T. & Langhammer, C. **Advanced Materials** 23, 4409-4414 (2011) - and ensemble - F. Sterl, et al., **Nano Letters** 2015, 15, 7949), and Y (ensemble - N. Strohfeldt, et al., **Nano Letters** 2014, 14, 1140.) In view of these results and the fact that, as we demonstrated here, our method allows plasmonic nanospectroscopy of single Pd nanoparticles, which are very poor scatterers, it becomes quite obvious that our approach is not restricted to Pd but can be applied to any metal hydride former. To give a specific example, we include here three optical p-C isotherms obtained for three individual Pd₉₀Au₁₀, as well as Pd₇₀Au₃₀ alloy nanoparticles, which are part of ongoing work on this alloy system.*

Q3: Furthermore, apart from a demonstration of the techniques, the scientific claims are rather speculative, e.g., "suggest direct correlation between length and type of grain boundaries and hydride-formation pressure"; "The observed structure-insensitivity of the hydride decomposition, which we also have observed for single crystalline nanoparticles, suggests a different phase transition pathway, most likely via an incoherent unloading process involving plastic deformation, as recently proposed by Griessen et al." As the authors claim to have evidence for a quantitative correlation, a quantitative model should then corroborate this statement. It is not clear to me, why plastic deformation affect only the hydride formation.

Our response: As for the "speculative conclusions" we may explain that these are phrased the way they are because we have the humble attitude that there is never 100% certainty and thus one should always have some reservations. Furthermore we indeed present a quantitative model for the grain boundary strain effect responsible for the observed grain-boundary length dependence of the hydrogenation pressure. The model predicts the reduction of the plateau pressure in absolute terms and in good agreement with the experimental data.

As for the second point, why plastic deformation mainly should affect hydride decomposition (not formation as stated by the reviewer), we argue that in general dislocation formation may take place in the core and shell (here we refer to a core or shell of hydride/metal, respectively) of nanoparticles or grains inside nanoparticles. Whether or not it really occurs depends on the thermodynamics and kinetics of the dislocation formation process. Customarily, dislocation theory is focused on energetic aspects of the dislocation formation, i.e., on thermodynamics. Referring to thermodynamics, one can indeed expect that the dislocations influence both branches of the adsorption

isotherms. The important point is, however, that the dislocation formation includes appreciable rearrangement of many metal atoms, and accordingly the kinetic aspects of the dislocation formation are expected to be equally or even more important, especially at the relatively low temperatures in the present case. More specifically, the metal-mass transport is expected to be quite different in the metal and hydride phase, due to e.g. spatial constraints induced for the transport by the presence of hydrogen atoms in the hydride phase. For this reason, the formation of dislocations may indeed be more likely during one of the branches of the sorption isotherms. At present, unfortunately, the understanding of the kinetic aspects of the dislocation formation in general and especially in nanoparticles with grains and during hydride formation/decomposition is very limited. Under such circumstances, extensive speculations focused on this subject may easily be misleading, and accordingly we do chose to not follow this way. In contrast, we hope and anticipate that our work will initiate theoretical studies of related aspects, which are far from trivial. With these reservations, we agree that the question raised by the Reviewer is important and that it should have been addressed more explicitly in our first version of the text. In the revised version, we added the key points of our explanations above to the main text on page 9 as follows:

Therefore, we argue that the reason for the asymmetry is related to not only the thermodynamics but, even more importantly at the present relatively low temperatures, to the kinetics of dislocation formation. For example, the metal atom mass-transport during plastic deformation is expected to be different in the metal and hydride phases, due to, for example, spatial constraints induced by the presence of hydrogen atoms in the hydride phase and/or the difference in spatial localization of the hydride-gas, metal-gas, and hydride-metal interfaces. Consequently, dislocation formation is expected to be governed by different kinetics during hydride formation and decomposition, constituting a reason for the observed asymmetry. However, at present the understanding dislocation formation kinetics in general, and especially in nanoparticles and during hydride formation/decomposition, is very limited (*J.N. Clark, J., et al., Three-dimensional imaging of dislocation propagation during crystal growth and dissolution. Nature Materials 14 (2015) 780-785. L.Y. Chen et al., Measuring surface dislocation nucleation in defect-scarce nanostructures. Nature Materials 14 (2015) 707-713. J.A. El-Awady, Unravelling the physics of size-dependent dislocation-mediated plasticity. Nature Commun. 6 (2014) 5926.*), preventing a more rigorous and quantitative analysis beyond the recent work by Griessen et al., which is in good agreement with our data.

Q4: Were the particles cycled? If not, the explanation may just be due to the initial barrier of defect formation.

Our response: Yes, particle cycling has no effect on the observed asymmetry between hydride formation and decomposition. This

becomes clear from Figure 4, where data sets 1-4 are all on the same sample and were measured in sequence (and thus cycled).

Q5: Finally, this study focuses on the hydrogen uptake by nano-particles. It is well known that in particular nanoparticles are sensitive to surface contamination. Although Pd is a relatively inert material, surface and also grain boundaries may be contaminated by carbon, and/or oxygen. In many materials, the stability of materials depends on the stability of the grain boundaries potentially weakened by contaminations (e.g., sulfur in steel).

*Our response: These nanoparticles are grown under high vacuum conditions at a base pressure of $< 5 * 10^{-7}$ Torr from a high purity Pd target (we have added this number to the methods section). Thus oxidized grain boundaries as well as carbon contamination is highly unlikely.*

Q6: In addition to the specific critics I miss some lines on future perspectives in the field. Pd hydride nano-particles are one of the few systems, where such investigations are possible. If the authors see further potential of this method, this is the place to mention it!

Our response: We have expanded the outlook part of the conclusion section that it now reads as:

*In a wider perspective, our general approach can be used to scrutinize the role of grains and grain boundaries in basically any metal hydride system based on the fact that numerous plasmonic sensing studies on ensembles of different hydride forming metal nanoparticle systems already exist (e.g. AuPd alloys⁴⁷, Mg^{48,49}, Y⁵⁰). Furthermore, it can be easily expanded to other processes of interest in metallic nanostructures where oxidation and reduction are a prominent example. Due to sizeable mismatch of the lattice spacing between metal and oxide, the formation of grains in the oxide is nearly inevitable and has long been expected to play a key role in oxidation/reduction processes (see, e.g., V.P. Zhdanov, *Oxidation of metal nanoparticles with the grain growth in the oxide. Chem. Phys. Lett.* 674 (2017) 136-140 and references therein). The underlying physics is, however, still far from clear, especially on the nm scale.*

Reviewer #2 (Remarks to the Author):

Overall, I am impressed by the both the quantity and depth of the experimental results. The investigation of the role of grain boundaries in the hydriding phase transformation is an open question and important for fundamental and technical reasons. However, I do have some concerns. I separated the concerns into major and minor.

Major

Q1: All of the information is spatially averaged over the entire particle. This needs to be explicitly stated in the text, especially when comparing it to refs. 13, 16 where the hydride formation is spatially mapped within the particle. It also assumes that the grains are columnar, i.e. extend through the thickness of the nanoparticles. Is there evidence for this? What could be the signature if the grains are not columnar?

Our response: To address the comment about spatial averaging we have added the following sentence on page 4 of the revised manuscript:

At the same time we also highlight that in plasmonic nanospectroscopy the obtained information is spatially averaged over the entire particle, in contrast to the recent EELS and X-ray studies where the hydride formation process can be spatially resolved inside a single nanoparticle.^{13, 16}

With respect to the grains being columnar or not, the fact that HRTEM imaging of the particles (Fig. S17) reveals lattice fringes is proof of columnar grains. To clarify this point we have reworded on page 9 to:

Furthermore, high-resolution TEM images reveal lattice fringes for each nanoparticle (Figure S17), in agreement with columnar grains stretching from the substrate through the entire particle.

Q2: There is no sensitivity to intragrain defects (particularly dislocations). It's known that dislocations play a very important role in the transformation pressure. Have the authors thought about how to exclude or account for the potential impact of dislocations within the grains?

Our response: This is indeed an important point raised by the Reviewer and as a first part of our response, we refer to our reply to Q3 by Reviewer 1. In addition we may add that, as seen in Figure 6c, our particles have grain radii on the order of 45 nm or significantly less. This means that each grain, which can be regarded as a single crystallite, is significantly smaller than the critical size for dislocation formation identified in the recent work by Ulvestad et al. (Three-dimensional imaging of dislocation dynamics during the hydriding

phase transformation, Nature Materials 2017, advance online publication). Thus, in this respect is likely that no dislocations will form inside the grain. Moreover, due to the presence of grain boundaries, which (in particular HAGBs) are highly defectuous, eventual defect-mediated contributions to the phase transformation are expected to take place at the grain boundaries. For clarity we have added the following text to the revised manuscript:

Finally, we mention that a competing contribution of dislocation formation inside the individual (single-crystalline) grains as mediator for the observed variations of hydrogenation pressure is highly unlikely in view of the fact that our grains, with grain radii of 45 nm or below (Figure 6c), are significantly smaller than the critical size for dislocation formation identified by Ulvestad et al.¹⁶

Q3: The data in Fig. 6 a-d shows basically no dependence of the transformation pressure on the independent variable being plotted if the initial and final points are excluded. For example, in Fig. 6b, from 200-800 nm the transformation pressure is effectively flat. I think the authors need to revise their conclusions a bit here. It's not clear to me that there is a strong effect. Fig. 6f does a bit better job. So, it might be that the different types of grain boundaries are complicating their analysis. Still, this is only mentioned at the very end and needs to be discussed more thoroughly.

Our response: We agree that the initial points are very important to reveal the magnitude of the dependence. However, we also stress that it is in this regime where the effect is expected to be strongest, due to the fact that the grain boundary length immediately attains a significant value if we take the step from a single crystal to a polycrystal. Hence it is to be expected that the initial points are the most crucial ones. To slightly weaken our statement we have rephrased our conclusion to:

We thus conclude that tensile lattice strain induced by hydrogen absorbed near grain boundaries is **an important** mediator of the observed significant spread in hydride formation equilibrium pressure of polycrystalline nanoparticles of the same size and shape.

With respect to the different types of grains, the observation of a dependence on grain boundary type actually rather strengthens our conclusion that complicates it. This, as we explain in the text, because due to the different structure of HAGBs and twin boundaries, if our general conclusion is correct, such a grain boundary type dependence is expected.

Minor:

Our response: We appreciate very much that the Reviewer has taken his/her time to provide us with such detailed suggestions for improvement of the text.

Q4: Line 23. "nearly lacking" is not a good word choice. Perhaps "beginning to be explored"

Our response: Fixed as proposed by the reviewer.

Q5: Line 31-32. The technique is not sensitive to strain so "we identify tensile lattice strain" should be removed. A better sentence would be "The absorption pressure dependence we observe is consistent with tensile lattice strain..."

Our response: Fixed as proposed by the reviewer.

Q6: Line 35-37. In my understanding, the technique will work for metals that have phase transformations where the plasmon conduction band changes by a lot. What are some examples of other systems where this is true?

*Our response: In fact, to be able to monitor a phase transformation using plasmonic nanospectroscopy the required changes can be very minor. In this respect PdH is a system with very minor electronic changes as it forms a metallic hydride. Many hydride phase transformations, such as for example Magnesium hydride, induce much more significant changes such as complete metal-to-insulator transitions. Moreover, larger volume changes induced by a phase transformation can also give rise to sizable plasmonic effects since the LSPR is highly particle size and shape dependent. In other words, the technique is very well suited to study phase transformations where the electronic changes are minor. Examples of other hydride systems studies using LSPR are: Magnesium (Shegai, T. & Langhammer, C. *Advanced Materials* 23, 4409-4414 (2011); F. Sterl, et al., *Nano Letters* 2015, 15, 7949), Yttrium (N. Strohhfeldt, et al., *Nano Letters* 2014, 14, 1140), and PdAu alloys (e.g. *Nano Letters*, 15, 3563-3570 (2015))*

Q7: Line 67. References need to be added to the significant literature that exists for nano crystalline thin films, especially those dealing with nano crystalline palladium

Our response: We have added the following selection of referneces on the topic on page 4 of the revised manuscript (In such systems grain boundaries are expected to be of significant importance due to the relative abundance of grain boundary sites compared to bulk materials with larger grains.^{18, 25, 26, 27, 28, 29}):

1. Mütschele, T. & Kirchheim, R. Hydrogen as a probe for the average thickness of a grain boundary. *Scripta Metallurgica* **21**, 1101-1104, doi:http://dx.doi.org/10.1016/0036-9748(87)90258-4 (1987).
2. Mütschele, T. & Kirchheim, R. Segregation and diffusion of hydrogen in grain boundaries of palladium. *Scripta Metallurgica* **21**, 135-140, doi:http://dx.doi.org/10.1016/0036-9748(87)90423-6 (1987).
3. Eastman, J. A., Thompson, L. J. & Kestel, B. J. Narrowing of the palladium-hydrogen miscibility gap in nanocrystalline palladium. *Physical Review B* **48**, 84-92, doi:10.1103/PhysRevB.48.84 (1993).
4. Natter, H., Wettmann, B., Heisel, B. & Hempelmann, R. Hydrogen in nanocrystalline palladium. *Journal of Alloys and Compounds* **253–254**, 84-86, doi:https://doi.org/10.1016/S0925-8388(96)02922-2 (1997).
5. Weissmüller, J. & Lemier, C. Lattice constants of solid solution microstructures: The case of nanocrystalline Pd-H. *Physical Review Letters* **82**, 213-216, doi:10.1103/PhysRevLett.82.213 (1999).
6. Lemier, C. & Weissmüller, J. Grain boundary segregation, stress and stretch: Effects on hydrogen absorption in nanocrystalline palladium. *Acta Materialia* **55**, 1241-1254, doi:https://doi.org/10.1016/j.actamat.2006.09.030 (2007)

Q8: Line 71. The TEM image shows residual stuff around the particle. Is this Pd or something else?

Our response: This is Pd that gets redirected from its initial trajectory during the physical vapor deposition process by colliding with the edge of the nanofabrication mask. This is also explicitly mentioned in the caption of Figure 3 (The small "satellite" features around the nanoparticle are formed during the nanofabrication. Due to their small size, they do not contribute to the measured signal).

Q9: Line 71: "This is a significant advance compared to state of the art". I don't agree with how refs 13 and 16 are characterized. In Dionne et al. the hydride decomposition process is investigated (see Fig. 3 in "In situ detection of hydrogen-induced phase transitions in individual palladium nanocrystals). In Ulvestad et al., the decomposition process can be mapped as well. It is just that the diffraction signal is very complicated due to the high dislocation density in the particles that the authors did not do this. It is as simple as doing loading/unloading measurements with bulk ensemble x-ray diffraction if no imaging is performed.

Our response: We have reworded to:

This is an advance compared to the state of the art,^{13, 16, 21, 22, 24, 31, 32} where only sequential measurements of individual nanoparticles are possible, meaning that artifacts due to measurement-to-measurement variations cannot be avoided.

Q10: Line 73-74: "where a new experiment is necessary for each studied nanoparticle". This is not true. In Ulvestad et al. up to 25 particles were

measured sequentially at each hydrogen partial pressure. Once the measurement of all particles was finished, the pressure was incremented and all particles were measured at the next pressure. Using the technique of Dionne et al., the same sequential approach can be used. No new experiment is required to measure multiple particles.

*Our response: In our opinion measuring the particles sequentially at the same pressure is still a different experiment per particle because they are not measured at the same point in time and during the time period from the first to the last particle being measured changes may occur that cannot be tracked in a sequential measurement. In this sense our approach brings an improvement because the particles are measured in parallel and not in sequence. Please refer to the rewording of the corresponding section in response to **Q9**.*

Q11: Line 71-74: The main difference of this technique is the simultaneous measurement. But it should also be pointed out that this technique does not give spatially resolved information unlike refs 13 and 16.

*Our response: Indeed, this is the main and quite important difference. We have made this clear in response to the above comments **Q9** & **Q10**. Also the comment about spatial resolution we have already addressed in response to **Q1** above.*

Q12: Line 103. Are there problems with these windows breaking during the Cr deposition and removal?

Our response: There can be such problems indeed (very dependent on the sample) but it actually turned out to be more challenging to mount the samples in the TEM/SEM holder without breaking them. Looking forward, in fact, we now (as part of an ongoing project) have good indications that for TKD analysis it is not even necessary to remove the Cr layer.

Q13: Line 104. Is it possible to get CrH₂ forming under the experimental conditions?

Our response: No, it is not possible as for example discussed in the textbook by B. Baranowski, Hydrogen in Metals II: Application-Oriented Properties (eds Georg Alefeld & Johann Völkl) 157-200 (Springer Berlin Heidelberg, 1978).

“...At normal pressure and temperature conditions, metallic chromium absorbs negligible amounts of gaseous hydrogen. An extrapolation to concentrations 10^{-1} - 10^0 in atomic ratios n_H/n_{Cr} would require

enormously high activities of hydrogen. But as chromium undergoes a reconstructive transition during the hydride formation, one could eventually expect more realistic pressures sufficient for these purposes. Therefore, both the decomposition and formation of chromium hydride seemed to be possible at high pressures of gaseous hydrogen only. ...”

Fig. 4.13. Formation (■●) and decomposition (□◇○) curves for chromium hydride in a P_{H_2} , T plot [4.75]

Q14: Lines 109-113. How is it ensured that the same particle is imaged in all of these steps? Are there fiducial markers on the sample holders used?

Our response: The size of the electron-transparent membrane on TEM substrates used in this work is 150x150um. The particles are dispersed at a very low concentration (as shown in SI Figure S5) and create unique pattern in the region of the TEM window. Therefore, it is easy to find the same particles in DFSS setup and SEM by comparing corresponding images, since window region is clearly visible in both instruments. In TEM, where the field of view is smaller, one can rely on the corners of TEM windows as reference coordinates and create a “map” towards the particle of interest using navigation option in TEM software and correlating distances between membrane edges and particles in the pattern (using prior SEM or DFSS images of the sample as a guide).

Q15: Line 118. Spelling of localized is incorrect.

Our response: corrected

Q16: Line 119. Since the technique is sensitive only to the surface plasmon, isn't it possible that the hydrogen-rich surface layer that forms before the transformation pressure could give the false signal that the entire particle is in the hydrogen-rich phase? The hydrogen-rich surface layer is well documented to form below the transformation pressure for the entire particle.

Our response: This is an interesting question, which we can address in the following way. First of all, our FDTD simulations (both for the present work as well as our earlier study on a different heterodimer arrangement with Pd nanocubes attached to Au nanoparticle plasmonic probe – Syrenova et al., Nature Materials 2015, 14, 1236.) were done based on the assumption that the entire Pd particle hydrogenates. Since the experimentally measured response and the FDTD-simulations in both studies are in good agreement, this can be seen as a first very strong indication that what we measure is not only a hydrogen-rich surface layer. The second argument is that the plasmon is sensitive to the whole volume and not only to the surface, as for example shown by first principle calculations on the specific example of Pd nanoparticle hydrogenation (M. Ameen Poyli, V. M. Silkin, I. P. Chernov, P. M. Echenique, R. Diez Muino, J. Aizpurua, Journal of Physical Chemistry Letters 2012, 3, 2556–2561.) Finally, based on FDTD computations for a different and ongoing project we can estimate the expected magnitude of a peak shift resulting from a hydrogen-rich surface layer. Assuming a 1 nm thick subsurface layer with stoichiometry PdH₁ (compared to PdH_{0.66} for bulk), we expect that the resonance shift for only a subsurface change would be 10-20% of the nominal shift, so between 10 and 5 times smaller than what we observe.

Q17: Line 131. I think it needs to be said that this information is spatially-averaged

Our response: We have already stated that the information is spatially average higher up in the revised text based on our response to Q1 and thus think it is not necessary to explicitly state it again here.

Q18: Line 145. What are the error bars in Fig. 3c?

Our response: We are not sure which error bars the Reviewer is referring to.

Q19: Line 166. How is equilibrium defined? How long is spent at each hydrogen partial pressure?

Our response: Typical time scales of the sample equilibration during absorption steps never exceeded 5-10 minutes, and during desorption

steps 5-20 minutes. Longer dwell times were chosen for steps around typical absorption and desorption pressures, to make sure that the signal doesn't change further. This information is presented in the SI where we show the individual hydrogenation traces for each measured particle (Figures S6-S10).

Q20: Line 166:168. Wording here is confusing. Several particles with different plateau pressures are equivalent to multiple plateaus for a single particle?

Our response: Just from the isotherm measurement alone, yes, one cannot tell the difference if the plateaus stem from several particles localized closely together within the same optically diffraction limited spot (hence in plasmonic nanospectroscopy they appear as one optical point source and thus as "one" particle) or from one particle with different plateau pressures. However, the corresponding SEM images then tell which of the two cases it is and allow us to interpret the isotherms correctly.

Q21: Line 181. Why do the authors think the particles are strongly adhered to the support?

Our response: Because, otherwise, they are expected to eventually peel off the surface due to the volume expansion/contraction upon hydrogenation. That said, of course it can be debated what "strongly" here means in absolute terms. Hence we have removed the word "strongly" in the revised text.

Q22: Line 190. "scrutinize" is not the right word choice. Suggest "investigate"

Our response: Fixed

Q23: Line 194. What is meant by lattice fringes here?

Our response: A lattice fringe is a periodic fringe in a TEM image, which is formed by the following two waves: (1) a transmitted wave exiting from a crystal; and (2) a diffracted wave from one specific set of lattice planes within the crystal. The spacing of the fringe corresponds to that of the particular lattice plane, and is therefore a fingerprint of the orientation of that particular (part of) the crystal.

Q24: Line 221-222. The factors are significant from 1-1000 nm for the loading pressure (Griessen Fig. 3). 1-100 nm for the unloading pressure.

Our response: Corrected and reworded to:

These factors are significant for particles in the 1-1000 nm regime, very much depending on the specific effect.

Q25: Line 226. Add reference to "Narrowing of the palladium-hydrogen miscibility gap in nano crystalline palladium" by J. A. Eastman, L. J. Thompson, and B. J. Kestrel Phys Rev B 1993

Our response: The reference has been added.

Q26: Line 245-247. What parameters are being fit in the model and how do the values compare to the fitted parameters of the Griessen model?

Our response: We use the standard and widely accepted parameters for hydrogen in Pd. For the thickness, l , of one side of the boundary, we employ a physically reasonable value, which cannot be considered as a fitting parameter. Thus, basically, we are not fitting any parameters but the calculations are done for the relevant temperature of 303 K using the numbers given in the SI.

Q27: Line 253-256. This seems like a strong conclusion given Fig. 6a-d showing a very weak dependence if the initial and final points are excluded.

*Our response: We refer to our response related to the same issue given above in the context of **Q3** by the same Reviewer (essentially the same comment).*

Q28: Line 474. What is "TM" polarization?

Our response: Transverse-Magnetic polarization. We have clarified this in the revised text.

Q29: Fig. 2 c-d. Why is there a mismatch in the peak locations for the computed and the measured figures? For example, in Fig. 2c the red curve has a maximum at about 440 nm, whereas the maximum in Fig. 2d is 490 nm.

*Our response: The reason is that the particle geometry chosen for the simulations is based on the **general** template used for fabrication. However, at the individual nanoparticle level there is size dispersion and as a result there will be variability in the peak positions. Also, after annealing the particle usually changes shape slightly (smaller and taller, more rounded). These effects make it basically impossible to*

perfectly match experiment and theory. Furthermore, the specific dielectric function used as input for the simulations also significantly affects the outcome. To this end, looking into the literature for available dielectric functions of metals reveals great variation for nominally the same system, which gives rise to quite significant uncertainty as it is not obvious which data set is “the correct one”. Hence the presented simulation example is rather here as a demonstration of the nature and magnitude of the response than aiming at the exact match between experiment and theory.

Q30: Fig. 3c: Why is there a positive shift in $\Delta\lambda$ right before the loading transformation? It is more noticeable in the blue curve than in the red. This can also be seen in many of the other isotherms, for example Fig. S12.

Our response: This is a good question and we do not know for sure. Speculatively, this could be attributed to the formation of the surface layer mentioned above, which forms at lower hydrogen pressure compared to the bulk transformation. At the same time, it can also be an artifact related to slight movement of the particle in the microscope and thus illuminating different pixels of the CCD, which in turn can give rise to the seen effect. However, we have no real proof for any of these possible explanations and for this reason we preferred to not mention it in the text. In particular because it does not affect our conclusions.

Q31: Fig. 4: In my understanding, this plot is to show that there is a big variance in the loading pressure but not in the unloading pressure. Less data could be used to make this point and make the figure potentially easier to read.

Our response: It is part of the point with this figure, indeed. A second important aspect of this figure is to show the number of data points that can be obtained per experiment and that it is possible to obtain a statistically significant set of single particle data with a reasonable amount of experiments. For this reason, we prefer to keep the figure as it is.

Q32: Line 516. These are not "crosses" they are "xs". A cross is like this +

Our response: Well, in our opinion what the reviewer proposes to be a “cross” is a “plus” so we think there is different definitions of “crosses”. Since there is only one type of “cross” symbol used in the figures, irrespective of the nomenclature used, it will not complicate interpretation of figures and we choose to leave it as it is 😊.

Minor comments on the SI:

Q33: Line 126-128. It's confusing to read that the simulation contains only the Cr mirror but then shows that the Cr mirror enhances the Pd signal as said in Line 105-107 in the main text.

Our response: We are not sure if we understand this question. All simulations referred to in lines 126-128 contain the Pd nanoparticle, in some case with and in some cases without the Cr layer in order to elucidate its role. In line 126 we explicitly state that we compare the total cross sections of the simulated Pd particle WITH and WITHOUT the Cr mirror to see how the Cr mirror affects the optical properties of the Pd scattering signal. The conclusion of these lines (126-128) is that the longitudinal mode (normal to the surface of the mirror and silicon nitride support) is not visible at all and so the simulations give an accurate description even for the simpler to use normal illumination. The only calculations without the Cr mirror and the Pd disk are presented in Figure S2d, where we calculate the efficiency of light coupling to the Pd particle. This further confirms the drawn conclusions.

Q34: Line 199. I do not see from the plots that there is a lack of a clearly defined, narrow LSPR. This needs a bit more explanation.

Our response: Here we probably chose a confusing wording. There is of course a clearly defined LSPR, the resonance is, however, spectrally quite broad (much broader as, for example, for Ag and Au for the reasons discussed in I. Zoric, et al., ACS Nano 2011, 5, 2535). So, what we mean is that the LSPR, as quantified by the amplitude of the enhanced electric field, for this system is relatively weak and spectrally quite broad. Hence we have reworded the confusing section to:

In both cases, the LSPR for this system is relatively weak and spectrally broad, as the electric field is enhanced only by a factor of 3, in agreement with the strong damping of plasmonic excitations in Pd¹⁸.

Reviewer #3 (Remarks to the Author):

This communication reports on an original combination of in-situ and ex-situ techniques for the study of hydriding equilibrium of Pd nanoparticles, together with microstructural characterisation. The key aspect of this work is the use of a single TEM window for both in-situ (plasmonic nanospectroscopy) and ex-situ (TKD and TEM) analyses, enabling to identify and track multiple particles submitted to the same experimental conditions. The results bring an interesting picture of the hydriding mechanism of palladium, because they provide the missing link between macroscopic ensembles of crystallites - i.e. from the well-known bulk Pd to nanostructured materials such as thin films and nanoparticles - and single particles/crystallites. Depending on the microstructure of each individual particle - combined with the p-C isotherms measured by plasmonic nanospectroscopy - interesting conclusions were drawn regarding the impact of grain boundaries on the miscibility gap between the hydride α and β phases. Therefore, I think that this paper is suitable for publication in Nature Communications. However, I have some comments, which should be addressed, notably concerning the decoupling of grain boundary effects from these of other defects, and the experimental error on the p-C isotherms.

Important general comments:

Q1: Reading this manuscript gives the impression that the authors measured the effect of the sole grain boundaries on the p-C isotherms, as if annealing the specimens up to 470°C did not affect other crystalline defects. Every crystalline defect has its distinct effect on the hydriding mechanism (see e.g. A. Pundt and R. Kirchheim, *Annu Rev Mater Res* 36 (2006) 555), and annealing activates defect recovery, not only from grain boundaries, but also from dislocations, vacancies, twins, impurities, etc. In my view, this should not be overlooked, especially when studying such nanomaterials with high defect densities. I recognise that, besides grain boundaries, the authors take twin boundaries as well into consideration in their analysis, but I am particularly curious about the dislocation density before and after annealing. Do the authors have such defect statistics already? Otherwise, could they measure it or extract it from their TEM data? If grain boundary effects are really overwhelming the other effects, this should be put into numbers. Otherwise, the authors should clarify why they think other effects can be neglected, and/or state which hypotheses were made regarding the effect of other defects.

*Our response: The point raised here by the reviewer is indeed relevant and related to a similar comments made by Reviewer #1 & 2 (see **Q3** of Reviewer #1 and **Q2** of Reviewer #2) and we thus also refer to our replies to their comments. To specifically answer the questions asked here, for the dislocation density before and after annealing, we do not have such information and numbers. However, we note that, except for*

the data presented in Figure 2 (which is not the data set we build our main conclusions on), we have only measured particles, which have been annealed and thus are expected to be very similar in terms of their defect structure and to what extent defects have been recovered. The only main difference between particles is their grain structure, which is the main argument for us to postulate that it indeed is the grain structure that mediates their properties. We also note that all our grains are significantly smaller than the critical size for dislocation formation during hydride formation in single crystalline Pd nanoparticles identified by Ulvestad et al. (Three-dimensional imaging of dislocation dynamics during the hydriding phase transformation, Nature Materials 2017, advance online publication), which makes dislocation formation during hydrogenation a very unlikely mediator for the observed effects.

Q2: The same remark applies to surface sites. In Pd, surface sites are by far the most energetically favoured, and at room temperature, surface coverage is close to unity even at low H₂ pressure (see e.g. M. Johansson et al., Surf. Sci. 604 (2010) 718-729). I am puzzled when I read the argumentation on p. 10 lines 227-237, I am not sure to understand what the authors mean here. In particular, the sentence “In contrast, the number of energetically favorable sites (per unit area) for hydrogen at grain boundaries, and thus their relative importance, is somewhat larger (due to the specifics of their structure) than at the nanoparticle surface)” makes little sense to me, because even if this is true, surface sites are filled in priority and close to saturation. This passage should definitely be reformulated for more clarity, and, in relation to previous comment, the idea that the effect of grain boundaries is the main effect observed here should be put into numbers (e.g. the double occurrence of the word “somewhat” in this paragraph is not convincing).

Our response: When writing "the number of energetically favorable sites (per unit area) for hydrogen at grain boundaries, and thus their relative importance, is somewhat larger (due to the specifics of their structure) than at the nanoparticle surface)", we had in mind that the surface of these nanoparticles is predominantly terminated by (111) facets while on grain surfaces more open facets are more abundant. This point has been now been explicitly declared in the revised text by rewording to:

In contrast, the number of energetically favorable sites (per unit area) for hydrogen at grain boundaries, and thus their relative importance, is larger (due to the specifics of their structure⁴⁶) than at the nanoparticle surface in view the fact that the surface of our nanoparticles are predominantly terminated by (111) facets, while on grain surfaces more open facets are more abundant.

We have also removed the terms “somewhat”, which in fact were not intended to insinuate that we are uncertain about our argumentation but the consequence of the wrong use of this word to express that the

difference is of the discussed parameters is not extreme.

To also address the relative importance of surface sites and grain boundary sites in our system, we highlight that the total grain surface area inside a polycrystalline nanoparticle with multiple grains is significantly larger than the external surface area of the particles. To explicitly mention this in the revised text, we have added the following sentence:

Moreover, the internal grain surface area in a polycrystalline nanoparticle is significantly larger than the external surface area, further highlighting the significance of grain boundary strain.

Q3: I have some interrogations regarding the experimental error related to the p-C isotherm measurements. I am surprised by the sentence on p. 7, lines 144-145 (“The particle composed of six grains shows almost identical pressures for hydride formation and decomposition”, see also caption of Fig. 3c). With such an important microstructural change, one would expect a sloped and narrowed miscibility gap after annealing, as correctly noticed by the authors when they refer to Refs. 18 and 41 in their manuscript.

Our response: With respect to slope we argue that the plateau indeed is sloped as it contains several (5-6) data points. Compared to isotherms of single crystalline nanocubes that we have measured in the same experimental (cf. S. Syrenova, et al. Nature Mater. 2015, 14, 1236.) the slope in the present case is much larger. The latter further corroborates the validity of our measurements in general (with respect to Q4 just below), that is, that the features we resolve are real and no artifacts due to lack of resolution along the pressure axis.

*With respect to the narrowing of the miscibility gap, we would argue that it actually should widen after annealing due to the observed significant increase in grain size (equaling “less nanocrystallinity”). This effect can actually be seen in figure 3f, where the isotherm is fully reversible for the measurement before and after the anneal. In this context, we also recall that according to the general theory the maximum miscibility gap is expected to be observed in the ideal macroscopic single crystal (see the discussion in our previous article [S. Syrenova et al. Nature Mater. (2015); Ref. 22]). Please also see our response to **Q6** below.*

Q4: Also, the double plateau feature identified on Fig. 3f is not very clear, at least by looking at the figure (it is more obvious on other isotherms though). The plateau split is identified at 37 and 42 mbar (p. 7, line 149). These values are quite close together, and make me wonder if they can really be resolved.

Our response: The double-plateau feature becomes very clear when

consulting the corresponding raw data that is the basis for Figure 3f and shown in the Supporting information Figure S10. In our opinion, and in view of the long equilibration times for each pressure step (cf. response to Q19 by Reviewer # 2), there is no doubt that this is a real effect (by the way, such double plateaus have been observed before also in other Pd nanoparticles, with the same asymmetry between absorption and desorption (C. Langhammer et al., The Journal of Physical Chemistry C 2012, 116, 21201).

Q5: The way the data points are spread in Fig. 4 - although the contrast between the absorption and desorption isotherms is very interesting - also makes me wonder if there is no artifact here, e.g. a higher error at higher pressures (plus the error bars on this graph are not real error bars). The authors should address in numbers the error on the plateau pressures and on the $\Delta\lambda_{\text{norm}}$ parameter derived from plasmonic nanospectroscopy. Maybe this could explain some unexpected results, e.g. why narrowed miscibility gaps cannot be resolved (there is indeed some scattering at the plateau borders on several isotherms in Figs. 4 and 5), and give more confidence in the author's analysis.

*Our response: With respect to the possible error on plateau pressures, as we explicitly write in the caption for figure 4, "The error bars (or small dots for desorption) correspond to the obtained width of the plateau along the pressure axis. However, in cases where the plateau spans directly between two data points, the true plateau width is expected to be lower and thus not resolved in our experiment (the single crystalline particle s2p5 is a good example)". So, indeed, they are no error bars but indicate the upper and lower bounds of the plateau and thus its width along the pressure axis. The mean value between this upper and lower bound is then defined as the "plateau pressure" as indicated by the symbol. However, this is not the main point here because there is no error of importance when comparing particles **within a data set** because, within this set, all particles were truly measured simultaneously and thus experience the exact same pressure. This is the key point that becomes possible with our method and constitutes the important step beyond state of the art (cf. also Q9-11 of Reviewer 2 and our corresponding response). In other words, within each data set, even if the absolute pressures have a certain uncertainty defined by the accuracy of the mass flow controller, the relative positions and widths of the plateau observed for the particles within this set are not affected by this error and thus allow the rigorous conclusions that we draw. This is the main point with Figure 4 and our experimental approach.*

When it comes to the question about narrowed miscibility gaps, we interpret the reviewer question as that s/he is wondering why we cannot resolve a clear narrowing of the coexistence region as, for example, function of grain boundary length. This is indeed a relevant point, which we address in the following way. First of all, as the Reviewer correctly points out, the lack of

conclusive correlation between grain structure and width of the miscibility gap is related to the uncertainty of the $\Delta\lambda$ parameter. As can be seen from the raw data hydrogenation traces in the SI, this number varies from particle to particle due to effects like scattering intensity (weaker scattering from a particle will yield more noise and thus a larger uncertainty for the $\Delta\lambda$ parameter) and is on the order of several nm. This is significantly larger than in our previous study (S. Syrenova, et al., Nature Materials 2015, 14, 1236) where we used Au nanoantennas as probes of attached Pd nanocrystals due to the significantly sharper LSPR peak for Au compared to Pd due to interband-damping in the latter (I. Zoric et al., ACS Nano 2011, 5, 2535). Consequently, the resolution is simply not high enough to conclusively show a correlation between grain boundary length and width of the miscibility gap in the regime where we have between 1 and 15 grains in a particle. However, as pointed out already in our response to Q3 above, for particle s5p2 shown in Figure 3f, where the change in microstructure from the before to after annealing state is much more drastic, there is a quite clear indication of a widening of the miscibility gap after annealing when the sample is comprised of two grains only. To address this point, we have added the following sentences to the revised manuscript:

Another aspect of grain boundaries that has been observed in nanocrystalline films^{16,23} is a characteristic narrowing of the miscibility gap. Inspection of our data in this respect does not reveal a significant correlation between grain boundary length and width of the miscibility gap (data not shown). We argue that the reason is the uncertainty of the $\Delta\lambda$ readout parameter, which is on the order of a few nm in the present case, and caused by the spectrally broad peak of the LSPR in Pd due to interband-damping.⁵⁰ This resolution is not enough to resolve this effect in the present regime of particles being comprised of 1-15 grains, where it is not expected to be very pronounced. However, we also note that, for particle s5p2 (Figure 3f), where the change in microstructure from the before to after annealing state is much more drastic, there is a quite clear indication of a widening of the miscibility gap after annealing, when the sample is comprised of two grains only.

Specific comments:

Q6: Page 2, lines 22-23: “In nanomaterials, however, investigations of grain boundaries are very challenging and nearly lacking”. I would remove “nearly lacking”, I think this statement is too strong. This is indeed challenging, but more than just several studies can be found on materials exhibiting very similar microstructures in terms of grain size, twin and dislocation densities (especially thin films).

Our response: We have reworded to:

In nanomaterials, however, investigations of grain boundaries are very challenging and just in the beginning of being explored.

Q7: Page 2, line 27: Why are the p-C isotherm measurements limited to 10 simultaneous particles? This is a key limitation that should be explained in the experimental section.

Our response: This is an important point because, in fact, there is no strict limitation to 10 nanoparticles since there is different ways to address significantly more by employing strategies like hyperspectral imaging. We have therefore added the following text to the Methods section:

This limits the maximal number of particles possible to analyze simultaneously to something between 10 and 25 (the higher number could be achieved by using electron-beam lithography to nanofabricate particles aligned in a single row). However, using concepts like hyperspectral imaging, significantly more particles can be analyzed simultaneously at the cost of significantly decreased data acquisition speed.^{55,56} (S. Chenet al., ACS Nano 2013, 7, 8824.; D. Zopf, et al., Biosensors and Bioelectronics 2016, 81, 287.)

Q8: Page 4, lines 67-75: The authors cite 7 references in a row and advertise their technique, claiming that both hydrogen absorption and desorption can be measured (let's call it claim A), and that more than 1 particle can be studied at the same time (claim B). It sounds like none of these references fulfill these claims, but it is not true (e.g. Ref. 21 makes claim A). The authors should separate this list of references between the ones that satisfy claims A and B in order to better isolate the innovative aspects of their work.

Our response: We agree with the reviewer and have addressed this point in response to questions Q9-Q11 by Reviewer #2 above and have reworded the corresponding section such that it now reads as:

This is an advance compared to the state of the art,^{13, 16, 21, 22, 24, 31, 32} where only sequential measurements of individual nanoparticles are possible, meaning that artifacts due to measurement-to-measurement variations cannot be avoided.

REVIEWERS' COMMENTS:

Reviewer #1 (Remarks to the Author):

As stated in my first review, the manuscript is a nice piece of work combining two interesting techniques aiming at insights into the hydrogen sorption behavior of nano-particles. The authors countered my critics in a convincing way. They added some missing information. However, from a point of understanding materials behavior, I am now more confused than before. I do not know how to suggest an easier (more concise) way either, so I recommend to publish the paper as is.

Reviewer #2 (Remarks to the Author):

All of my questions except for one have been answered. I would like to see error bars added to Fig. 3c and Fig. 3f. Currently there is no uncertainty indicated. Aside from this final request, I recommend publication of this nice piece of work.

Reviewer #3 (Remarks to the Author):

The comments I raised have been nicely addressed. The paper reads much better now, and its contribution to the state of the art, as well as its limitations, can be clearly identified by any reader. I think the paper can now be published as it is, and I thank the authors for this nice Piece of work.

Point-to-point response to referees

Reviewer #1

As stated in my first review, the manuscript is a nice piece of work combining two interesting techniques aiming at insights into the hydrogen sorption behavior of nano-particles. The authors countered my critics in a convincing way. They added some missing information. However, from a point of understanding materials behavior, I am now more confused than before. I do not know how to suggest an easier (more concise) way either, so I recommend to publish the paper as is.

OUR REPLY: We thank the Reviewer for her/his positive response.

Reviewer #2

All of my questions except for one have been answered. I would like to see error bars added to Fig. 3c and Fig. 3f. Currently there is no uncertainty indicated. Aside from this final request, I recommend publication of this nice piece of work.

OUR REPLY: We have added a different version of Figs. 3c and d to Supplementary Figure 10 (where the raw data traces also are shown), where error bars are included (see below). We opt to not put the version with error bars in the main text because we feel the figure will be too busy. In this way both versions will be available. We also refer to the version with error bars in the caption of Figure 3 in the main text.

Reviewer #3

The comments I raised have been nicely addressed. The paper reads much better now, and its contribution to the state of the art, as well as its limitations, can be clearly identified by any reader. I think the paper can now be published as it is, and I thank the authors for this nice Piece of work.

OUR REPLY: We thank the Reviewer for her/his positive response.